# Antifungal Properties of Essential Oils and Their Compounds for Application in Skin Fungal Infections: Conventional and Nonconventional Approaches

**DOI:** 10.3390/molecules26041093

**Published:** 2021-02-19

**Authors:** Aswir Abd Rashed, Devi-Nair Gunasegavan Rathi, Nor Atikah Husna Ahmad Nasir, Ahmad Zuhairi Abd Rahman

**Affiliations:** 1Nutrition, Metabolism and Cardiovascular Research Centre, Institute for Medical Research, National Institutes of Health, Ministry of Health Malaysia No.1, Jalan Setia Murni U13/52, Seksyen U13 Setia Alam, Shah Alam 40170, Malaysia; rathidevinair@moh.gov.my; 2Department of Biology, Faculty of Applied Science, Universiti Teknologi MARA (Perlis Branch), Arau 02600, Malaysia; atikah1388@uitm.edu.my; 3Cancer Research Centre, Institute for Medical Research, National Institutes of Health, Ministry of Health Malaysia, No.1, Jalan Setia Murni U13/52, Seksyen U13 Setia Alam, Shah Alam 40170, Malaysia

**Keywords:** skin fungus, essential oils, in vitro, in vivo, intervention

## Abstract

Essential oils (EOs) are known to have varying degrees of antimicrobial properties that are mainly due to the presence of bioactive compounds. These include antiviral, nematicidal, antifungal, insecticidal and antioxidant properties. This review highlights the potential of EOs and their compounds for application as antifungal agents for the treatment of skin diseases via conventional and nonconventional approaches. A search was conducted using three databases (Scopus, Web of Science, Google Scholar), and all relevant articles from the period of 2010–2020 that are freely available in English were extracted. In our findings, EOs with a high percentage of monoterpenes showed strong ability as potential antifungal agents. *Lavandula* sp., *Salvia* sp., *Thymus* sp., *Citrus* sp., and *Cymbopogon* sp. were among the various species found to show excellent antifungal properties against various skin diseases. Some researchers developed advanced formulations such as gel, semi-solid, and ointment bases to further evaluate the effectiveness of EOs as antifungal agents. To date, most studies on the application of EOs as antifungal agents were performed using in vitro techniques, and only a limited number pursued in vivo and intervention-based research.

## 1. Introduction

Fungi are ubiquitous environmental microorganisms that may be categorized, according to their dimorphic morphology, as unicellular (yeast) or filamentous (molds). Almost one million mycotic species have been reported to exist in nature, with approximately 200 species identified as human pathogenic [1]. It has been found in recent years that fungal infections have contributed to increased mortality rates [2]. This phenomenon has been linked to certain age groups, especially premature neonates, infants and elderly people who are susceptible to underdeveloped or poor immune systems [3,4,5]. The most common species associated with deadly invasive and superficial infections are *Candida* sp., *Aspergillus* sp., and *Cryptococcus* sp. [6]. In addition, *Fusarium* sp. has been shown to cause opportunistic invasive fungal infections [5,7].

*Aspergillus* spp. is a filamentous and ubiquitous fungi with *A. fumigatus* as the major species associated with human disease, followed by *A. flavus*, *A. niger* and *A. terreus* [3,8,9]. In addition to the most common species, several other emerging species exist, including *A. clavatus*, *A. nidulans*, *A. glaucus* and *A. ustus* [6,9]. *Fusarium* spp. are other fungi that can cause human infections, and are the primary cause of fungal keratitis. This fungus is the second most common to infect severely immunocompromised patients and cause disseminated infection [10]. *F. solani* has been identified as the most frequent pathogen in fusarial keratitis incidence, while *F. oxysporum* leads to major incidences of onychomycosis [11,12,13]. *Candida* spp. are tiny, oval-shaped fungi with a thin cell wall that are capable of budding or fission. Among the identified species, five are the leading cause of invasive infections (*C. albicans, C. glabrata, C. parapsilosis, C. tropicals* and, *C. krusei*) [14]. Invasive candidiasis often occurs as a form of healthcare-associated infection, where affected patients are typically receiving broad-spectrum antibiotic treatment, immunosuppressants, or suffering from cancer [15]. Candidiasis infections typically exist on the epithelial surfaces of the mouth, gastrointestinal tract, vagina and skin surfaces. *C. albicans* remains the most common cause of skin, nail and mucous membrane infections in healthy individuals, in whom it may also induce more severe infections of the vital organs [16,17]. 

In general, fungal diseases are differentiated into four groups: dermatophytosis, subcutaneous mycoses, systemic mycoses and other mycoses [4]. Dermatophytosis is caused by dermatophytes that attack and grow on dead animal keratin. *Epidermophyton, Microsporum* and *Trichophyton* are the three main genera related to dermatophytes. Dermatophytes are known as a species of fungi that typically infect and invade a living host’s skin, hair and nails. Diseases caused by dermatophytes are typically classified according to the infection site, but are broadly referred to as tinea. Several forms of tinea are common such as *Tinea capitis* (scalp and hair), *Tinea corporis* (nonhairy skin), *Tinea barbae* (beard), *Tinea cruris* (groin), *Tinea manuum* (hand), *Tinea pedis* (feet) and *Tinea unguium* (nails, also called onchomyosis) [18]. According to current practice, five classes of conventional antifungal treatments are commonly applied. Figure 1 shows each antifungal agent and its mechanism of action. 

However, the treatment of fungal infections has encountered serious difficulties in the form of increased resistance due to the extensive use of antifungal agents. This situation has led to the insight that alternative, nonconventional approaches are required for effective antifungal treatment strategies. One of the possible directions proposed is the use of essential oils (EOs) as potential antifungal agents. EOs can be extracted from various plant parts and are volatile, aromatic, concentrated, hydrophobic oily liquids. Monoterpenes, sesquiterpenes and phenolic compounds are the key components of EOs. Phenolic compounds are chiefly responsible for the antimicrobial properties of Eos [20]. Numerous studies have proven the efficacy of EOs in antifungal treatments, although not all have addressed the underlying mechanisms of action [21]. The most widely used parameter in the antimicrobial assessment is minimum inhibitory concentration (MIC) and minimum fungicidal concentration (MFC), i.e., the lowest antifungal agent concentration required to inhibit fungal growth or to kill mycetes, respectively [2,22]. At present, thyme (*Thymus* sp.), tea tree (*Melaleuca alternifolia*), peppermint (*Mentha piperita*) or clove oil (*Syzygium aromaticum*) are the most widely tested EOs in terms of their antifungal activities [23,24,25,26]. Nevertheless, examination of the potential of EOs has extended to other types of oils [27]. Several examples of the use of EOs in the treatment of fungal diseases are described below.

Thyme EO (*Thymus* sp.) is recognized as a promising antifungal agent due to the presence of thymol and carvacrol at high concentrations [28]. A study conducted with one of the most common thyme EO, *T. vulgaris*, indicated strong antifungal activity against *A. flavus* at a concentration of 350 ppm [29]. Rasooli and colleagues performed a comparative study of antifungal activity against *A. niger* using *T. eriocalyx* (Ronniger) Jalas, Rech.f. and *T. x-porlock*, showing superior activity of *T. eriocalyx*, with a MIC value of 125 ppm [30]. Another EO that has been widely explored is cinnamon oil (*Cinnamomum* sp.), with cinnamaldehyde as its major component. An investigation of the antifungal effect of *C. zeylanicum* exhibited a synergistic effect with Fluconazole against *A. fumigatus* [31,32,33]. Similarly, *Eucalyptus globulus* EO has been shown to possess potent antifungal activity; one study examined its effect on growth inhibition *of A. flavus, A. niger, A. terreus* and *A. fumigatus* [34]. *S. aromaticum* has also shown antifungal efficacy against different *Aspergillus* spp. (*A. terreus, A. flavus* and *A. fumigatus*) [29,35,36,37]. Additionally, *Citrus* sp., *Mentha* sp., and *Cuminum* sp. have been applied against *Aspergillus* spp. [36,38].

Fungal infections arising from *Candida* spp. are called candidiasis; the most commonly implicated species is *C. albicans*. Fluconazole and itraconazole are widely utilized for such infections, but these agents have caused azole resistance within *Candida* spp. [3,39,40]. The antifungal properties of EOs derived from *Melaleuca* sp., *Origanum* sp., *Thymus* sp., *Mentha* sp., *Syzgium* sp., *Coriandrum* sp., *Cuminum* sp. and several others against *Candida* spp. were noted [2]. *Melaleuca* sp. is the main genus to have been studied for its anticandida properties. In a study by Mondello and colleagues, it was observed that *C. krusei* and *C. gabrata* showed sensitivity towards *M. alternifolia* EO; additionally, all azole resistant *C. albicans* were killed within 30 and 60 min at 1% and 0.25% concentration, respectively [41]. Antifungal activity was also observed from *T. pulegiosides* EO, that inhibited *C. albicans*, *C. glabrata*, *C. parapsilosis*, *C. krusei* and *C. guillermondii* with a MIC value range of 0.32–0.64 μL/mL [28]. Peppermint EO is also highly capable as an antifungal agent. Studies have revealed its inhibition capacity against *Candida* spp., dermatophytes and *Aspergillus* spp. when tested within 40–7000, 800–3500 and 400–3500 μg/mL, respectively [42]. *Lavandula angustifolia*, or lavender EO has also exhibited excellent antifungal activity against *Candida* spp. [43,44]. *Cumin* EO effectiveness as an antifungal agent has also been proven with respect to *C. albicans*, resulting in MIC values of 3.90 and 11.71 μg/mL [45].

Despite the existence of numerous studies suggesting the suitability of EOs as an alternative treatment strategy for fungal infections, it should be noted that these properties are highly dependent on the composition of the EO itself. These compositions are prone to significant variability based upon the use of different plant parts or the harvesting season [27]. Terpenes and their metabolic derivatives constitute the major component of Eos; these include functional derivatives of alcohols (geraniol), ketones (menthone), esters (cedryl acetate) and phenols (thymol). In addition, nonterpene compounds derived from phenylpropane (eugenol) are present in smaller percentages [46]. Thus, the effects are subject to variations, due in part to combinations of ingredients. *Curcuma longa* L. EO effects have been demonstrated against *A. flavus*, in addition to aflatoxin inhibition. An interesting observation with exposure to this EO was reported via scanning electron microscopy (SEM) analysis, that signified hyphae membranes and conidiophores damage in *A. flavus* [47]. Likewise, EO of *Matricaria chamomilla* L. flower was investigated against *A. niger*, revealing apparent destruction of cytoplasmic membranes and intracellular organelles, plasma membrane detachment from the cell wall and complete disorientation of the hyphal compartment [48]. In addition to modification of the ultrastructure, a number of studies have examined the underlying mechanisms behind the proapoptotic effects of EOs. An example of this is *Ocimum sanctum* L. EO, which induced major cytotoxicity in *C. albicans*, where complete ergosterol depletion, membrane disintegration, DNA fragmentation, increased outsourcing of membrane phosphatidylserine and cytochrome c oxidase activity were observed [49].

However, it has also been suggested that the cytotoxic effects of EOs are mediated by induction of reactive oxygen species (ROS) over synthesis and oxidative stress. A study performed with *Anethum graveolens* L. seed EO revealed apoptosis in a *C. albicans* strain via ATPase activity decrease, chromatin condensation, DNA fragmentation, phosphatidylserine exposure, cytochrome c release and metacaspase activation. This study highlighted the role of L-cysteine in the prevention of apoptosis, which is a sign of ROS action [50]. Aflatoxins, i.e., harmful fungal toxins, have also attracted much interest relative to the potential effects of EOs [21]. Several studies have suggested positive inhibition outcomes for aflatoxin with the application of EOs, such as *Chenopodium ambrosioides* L., that suppressed the synthesis of aflatoxin B1 by the aflatoxigenic strain of *A. flavus* [51]. Similar results were also observed with *Zataria multiflora Boiss* EO, which decreased the growth of aflatoxin in *A. parasiticus*, indicating a link with gene inhibition of the biosynthesis pathways for aflatoxin [52].

To date, our knowledge of the mechanisms of action of EOs is still minimal, with emphasis on concentration-dependence. In general, some apparent effects of EOs include loss of membrane integrity, reduction of ergosterol levels, inhibition of wall formation, inhibition of gene expression, and suppression of membrane ATPases and cytokine interactions [53,54]. As such, the effectiveness of the antifungal characteristics is believed to be associated with the properties of the respective major component, but standardization of assessment methodologies and measurement units is important for further comparisons and investigations [27,55]. It is widely anticipated that EOs will eventually replace traditional agents. Considerable research has been performed and reported on the possible utilization of these nonconventional approaches, mostly in the form of ointment, gels and others [56,57].

In this review, we focus on the potential utilization of EOs as an alternative treatment of fungal infections in humans, along with the classification of these approaches as conventional and nonconventional techniques.

## 2. Results

The search resulted in a total of 192 articles from the three search engines: Pubmed (42 articles), WoS (61 articles) and Scopus (89 articles). Nevertheless, 86 articles were identified with a refined search based on the availability of full text, peer reviewed articles and library collection access. Upon further assessment, only 41 full articles were found to be relevant and were included for final review (Figure 2). All of these articles were printed for further assessment. Upon conducting a thorough review, these 41 articles were classified into three categories: 32 articles were categorized as in vitro research, 7 as combination of in vitro and in vivo research and 2 articles as clinical interventions. Out of this, 5 in vitro studies and 3 combination studies were classified as descriptions of nonconventional technology (Table 1). Due to a lack of randomized controlled trials (RCT) in all but two articles, we could only include this topic in the form of a narrative review.

## 3. Discussion

In general, body aches are recognized and accepted as an initial symptom that could signify various underlying ailments. Dermatitis is normally acknowledged as a common condition of skin irritation that manifests in the form of dry skin or rashes. It usually instigates itchiness and may also cause blisters or skin flaking. Common conditions include atopic dermatitis (eczema), seborrheic dermatitis (dandruff) and contact dermatitis [98,99]. Acne is another inflammatory condition widely known to affect teenagers. Although this is usually considered a common ailment, certain severe forms can result in scarring, that may affect self-esteem and social interaction, along with causing psychological distress [100].

Infections are typically defined as the acquisition of a microbe by a host, which results when the microorganism is not eradicated from the host upon direct contact, thus initiating disease [101,102]. It has been observed that fungal infections are becoming more prominent; this could be explained by the sharp growth of high-risk populations and the application of treatment modes that allow for improved survival rate [103]. Several variations of endemic fungal infections have been observed with specific geographical distribution due to climate change, human habitats expansion, ease of travel and population migrations [1]. There are four key prerequisite criteria for fungi to infect humans: its ability to colonize or penetrate surface barriers, its ability to absorb nutrients via lysis and absorption of human tissue, its ability to counteract innate and adaptive immunity associated pressure, and its ability to grow at typical human body temperature [104].

The current practice in the treatment of fungal infections is solely focused on the application of traditional antifungal agents. However, as new species of fungi emerge, different approaches are necessary, given the increased resistance toward commonly available antifungal medications. Based on extensive research performed on EOs, the potential of these oils for antifungal based therapies has gained huge attention in recent years. Our review highlights numerous studies that have evaluated the potential of various types of EOs against human pathogenic fungi that were examined via in vitro, in vivo as well as clinical research. The listed research articles were also defined and classified as conventional and nonconventional, where applicable.

### 3.1. EOs in Conventional Approaches

Among the in vitro studies, *Candida* spp. infections were among the most common. The hypothesis of the previous study was that the spread of candida infections could be avoided in the occurrence of a healthy immune system, except when scrapes or cuts are found on the body [105]. Numerous EOs were shown to be suitable to control candida infections. Commercially available lemon EOs have gained attention and also been shown to possess antifungal potential against three *Candida* species (*C. albicans*, *C. tropicalis* and *C. glabrata*), where the growth of *C. albicans* specifically was inhibited across the full spectrum of concentrations used [79]. In another study, anticandidal activities were examined against *C. torulopsis*, *C. stellatoidea* and *C. albicans* using *Physalis angulata* EO. *P. angulata* is an annual herbaceous plant in the Solanaceae family which is recognized for its medicinal value. *P. angulata* EO demonstrated excellent characteristics as an antifungal agent against all three candida strains that are usually resistant to antibiotics. As such, additional research on its anti-infective properties is warranted [72].

Cabral and colleagues evaluated the antifungal potential of *Juniperus communis* subsp. *alpina* (Suter) Celak needles EO [75]. *J. communis* L. is an evergreen shrub or tree with fleshy female cones, where the cone scales resemble berries; it occurs throughout Europe, Asia and North America [106]. The evaluation of juniper needle oil revealed excellent activity, in particular to *M. canis* and *T. rubrum*, with MIC and minimum lethal concentrations (MLC) of 0.32 μL/mL [75]. The oil was made up largely of monoterpene hydrocarbons (78.4%), with the main components being sabinene (26.2%), α-pinene (12.9%) and limonene (10.4%). Cell viability studies indicated the superiority of needle oil with no cytotoxic effects observed in HaCat keratinocytes when tested at 0.32 and 0.64 μL/mL, compared to a previous study on berry EO [75].

Another study investigated the effect of two EOs (*E. caryophylla* and *M. sp cf piperita*) on six human pathogenic fungi (four filamentous fungi (dermatophytes): *T. rubrum* (1 and 2), *T. violaceum*, and *T. soudanense*) and two pathogenic yeasts (*C. albicans* 1 and 2). *Mentha sp cf piperita* is a herbal plant belonging to the Lamiaceae family, native to temperate regions, with a fresh, sweet smell that is traditionally applied in the treatment of gastritis, muscular pains or toothache [107]. *E. caryophylla* (Myrtaceae) (clove) is a tree of 10–12 m in height that originates from Indonesia but which is also harvested in African countries. It is widely used as a spice, as well as for the treatment of toothache and in infection remedies [108]. The study showed that *Mentha sp cf piperita* EO presented weak activity, with MIC of 2.5 μL/mL against *Tricophyton strains*, while no activity was shown with *C. albicans* [74]. Additionally, *E. caryophylla* EO exhibited the highest activity, with MIC and MFC of 0.25 µL/mL and 0.125 µL/mL for filamentous fungi and MIC of 0.5 µL/mL for both yeast strains, while the MFC value was 1 µL/mL for one yeast strain and was not determined for the second [74].

In a study by Valente et al. [65], the potential of *D. carota subsp. gummifer* EO as an antifungal and anti-inflammatory was evaluated against multiple collections and clinical strains via macrodilution broth assays. Among the tested strains were two *Candida* strains isolated from recurrent cases of vulvovaginal candidosis (*C. krusei* H9 and *C. guillermondii* MAT 23), three type strains from the American Type Culture Collection (*C. albicans* ATCC 10231, *C. tropicalis* ATCC 13803, *C. parapsilosis* ATCC 90018), one type strain from the Colección Española de Cultivos Tipo (*C. neoformans* CECT 1078), three dermatophyte strains isolated from nails and skin (*T. mentagrophytes* FF7, *M. canis* FF1, *E. floccosum* FF9), two type strains from CECT (*T. rubrum* CECT 2794, *M. gypseum* CECT 2908), one *Aspergillus* strain isolated from bronchial secretions (*A. flavus* F44) and two type strains from ATCC (*A. niger* ATCC 16404 and *A. fumigatus* ATCC 46645). Chemical characterization of the oil revealed high amounts of monoterpenes with geranyl acetate and α-pinene as the major components. This EO action was shown to be active against dermatophytes and *C. neoformans* (MIC range: 0.32–0.64 μL/mL). The promising results shown by this EO led to the recommendation of future studies with a specific emphasis on peripheral and central nervous systems (CNS), implying the need for in vivo research focused in dermatophytosis and inflammatory disease management [65].

Two more genera of fungi are commonly related to skin diseases: *Microsporum* and *Trichophyton*. These opportunistic filamentous fungi belong to dermatophytes [109]. From our review, we found that many studies have been done to find methods to counter these two species. *Artemisia* is one of the most widely distributed species of the Asteraceae family, and is well-known for its pharmacological benefits. *A. sieberi* was tested on dermatophytes against several strains (*T. rubrum, T. mentagrophytes, M. canis, M. gypseum, T. schoenleinii* and *T. verrucosum var. album*), revealing higher sensitivity of *M. gypseum, T. rubrum* and *M. canis* and suggesting its suitability as a topical antifungal agent. However, this study indicated that collection site and harvest time affect the yield and chemical composition of the EO, although no significant differences were observed with respect to its antidermatophyte activities [59].

Rezgui and colleagues studied the effects of *M. vulgare* EO, extract and its active compound (marrubiin), and observed approximately 50% inhibition for *T. mentagrophytes* and *E. floccosum* with respect to marrubiin tested at 100 mg/mL. Antiphytopathogenic activity was also conducted, with only marrubiin being found to be active against *B. cinerea* at the highest dose (32.40%). The authors concluded that *M. vulgare* and marrubiin could be used as natural antifungal agents for skin dermatophyte infections [73]. *Myristica fragrans* is an edible plant that grows in the evergreen forests of West Africa. The seeds have both economic and medicinal value, especially for the local community. Inhibition was observed against several pathogenic fungi, e.g., *C. tropicalis* (1.3 cm), *C. albicans* (0.8 cm), *R. miehei* (0.6 cm) and *C. glabrata* (0.6 cm) with *M. fragrans* EO, although no inhibition was reported with *A. niger* and *A. fumigates* [61].

In Lebanon, Lamiaceae species such as *C. capitatus* L., *L. stoechas* L., *L. angustifolia* Mill., *M. spicata* L. subsp. condensata, *O. syriacum* L.,* R. officinalis*, *S. fruticosa* Miller., *S. cuneifolia* Ten., *S. thymbra* L., *T. spicata* L., and *V. agnus-castus* L. are quite popular due to their use as a food, a condiment or as traditional medicine in the treatment gastrointestinal disorders and microbial infections. The EOs of Lamiaceae sp. was shown to be efficient, especially toward *T. rubrum* and *C. albicans*, that reported MIC values ranging between 64–128 µg/mL; this was attributed to the presence of large amounts of thymol and carvacrol components [60]. A recent review by Karpinski highlighted the antifungal properties of EOs from 72 Lamiaceae plants. Linalool, β-caryophyllene, limonene, β-pinene, 1,8-cineole, carvacrol, α-pinene, p-cymene, γ-terpinene, and thymol were identified as the major components among the tested plants; these compounds are known to possess antifungal characteristics. On the whole, the review concluded that more than half of these EOs presented MIC activity (<1000 μg/mL), with Clinopodium, Lavandula, Mentha, Thymbra and Thymus EOs displaying the best activities [110].

In addition to the species mentioned above, Lebanon is also well-known for EOs from conifer plants such as *A. cilicica, C. sempervirens, J. excelsa, J. oxycedrus, C. libani* and *C. macrocarpa* gold crest. The efficacy of such oils was evaluated against *C. albicans* and several isolates of *Trichophyton* spp. Based on the results, the highest sensitivity was observed with *Trichophyton* spp. to all tested EOs (MIC: 32–512 μg/mL). This study, however, highlighted interesting activity of a *C. macrocarpa* EO on dermatophytes (MIC: 32–64 μg/mL), whereby each major component, as well as an artificial EO, were tested on *T. rubrum*, suggesting a possible contribution of a minor component to the overall observed activity. The findings strengthen the potential of EOs of Lebanese conifers for use in antimicrobial preparations for the treatment of superficial infections [67].

In another test against *T. rubrum* and *T. mentagrophytes*, the antifungal activities of *A. betulina* and *C. album* EOs was evaluated using electron microscopy based on the mechanism of action. It was found that fungal growth was inhibited on all plates exposed to the EOs volatiles with different inhibition rates observed among the various fungal species, EOs and tested volumes. The highest inhibition was recorded on *T. rubrum* at 40 μL via the action of *A. betulina* EO, with a fungal growth index (FGI) of 2.3% indicating its capacity as a strong antifungicidal agent. The hyphae and spores of *T. rubrum* were also destroyed due to the volatile effects. The major components identified in *A. betulina* EO are limonene (29.8%), menthone (21.6%), and isomenthone (14.7%), while pinene (27.4%) and myrcene (14.5%) makes up the major part of *C. album* [69]. In the formulation of skincare products, the *C. album* (Thunb) Bart. & H. L. Wendl (Rutaceae) has been used by the Khoisan people by rubbing it on their skin. However, it is less popular in South Africa as a traditional medicine. The mechanism of action of *A. betulina* EO volatiles was shown to be through spore growth inhibition as well as production and morphology alteration via destruction of hyphae and spores. Overall, this research gave remarkable insights into the possible use of *A. betulina* as an antifungal agent [70].

Another study on the potential of EOs from *L. luisieri* and *C. citratus* against *T. rubrum* and *T. mentagrophytes* was conducted, whereby strong antifungal activity was observed with most clinical strains. Positive interaction between *L. luisieri* EO combined with terbinafine was observed against terbinafine-resistant strain (Tr ATCC MYA-4438). Significant reduction of the germination was observed above 100 µg/mL. Both oils were found to be safe to macrophage mammalian cells at the tested concentrations. This study describes the antifungal activity of *L. luisieri* and *C. citratus* EOs against dermatophytes, which could be useful in the design of new formulations of topical treatments [77].

Since ancient times, several *Thymus* species have been used in traditional medicine to combat pathogenic microorganisms. In a study by Goncalves et al. [63], chemical profiling and antifungal activity of four types of oils from *Thymus zygis* subsp. *sylvestris* were evaluated against yeasts, dermatophyte and *Aspergillus* strains. Carvacrol (25.0%), thymol (23.8%), geranyl acetate (20.8%), geraniol (19.8%) and linalool (30.0%) were the major identified components. MTT (3-(4,5-dimethylthiazol-2-yl)-2,5-diphenyltetrazolium bromide) assay revealed that the oil composed of high carvacrol concentrations presented stronger antifungal activities against dermatophytes, with no cytotoxic effect at 0.08–0.16 μL/mL, for up to 24 h. A lack of cytotoxicity was also observed with this oil in mouse skin dendritic cells at concentrations where significant antifungal activity was obtained (0.16 μL/mL). Based on these findings, it was proposed that *T. zygis* EO is highly capable as an antifungal agent with minimal detrimental effects, and that it should be explored in more detail [63].

In combatting ringworm infections, *S. khuzistanica* EO, ethanol and aqueous extract were tested for their antifungal activity. In the study, the antifungal activities of *S. khuzistanica* EO (MIC: 40–190 µg/mL) were found to be higher than its ethanol extract (MIC: 40–770 µg/mL), while *S. khuzistanica* aqueous extract had no antifungal activity against dermatophytes (MIC: 1550–3100 µg/mL). This study noted the potential of *S. khuzistanica* EO as a topical therapy in the treatment of skin fungal infections. It also noted that both *S. khuzistanica* EO and its ethanol extract could be further explored for their antifungal activities [64].

*S. officinalis* L. belongs to the Lamiaceae family; it was evaluated for its antifungal and anti-inflammatory activity. *S. officinalis* EO indicated antifungal activities against dermatophytes and significantly inhibited nitric oxide (NO) production stimulated by lipopolysaccharide (LPS) in macrophages without affecting cell viability in concentrations up to 0.64 μL/mL. This study claimed to be the first report on the in vitro anti-inflammatory potential of *S. officinalis* oil; the authors demonstrated that the bioactive concentrations of *S. officinalis* oils did not affect the mammalian macrophages and keratinocytes, making them suitable for use in skin care formulation for cosmetic and pharmaceuticals applications [66].

The antifungal activity of *D. tenuifolium* EO was evaluated against yeast, dermatophyte and *Aspergillus* strains. The oils revealed significant antifungal activity against *C. neoformans* and dermatophyte strains, and significantly inhibited NO production stimulated by LPS in macrophages, without affecting cell viability at concentrations ranging from 0.64–1.25 L/mL. A chemical analysis found that the oil was mostly composed of monoterpene hydrocarbons with myrcene as the major component. This research provided significant information about the pharmacological activities of *D. tenuifolium* EO which is relevant to its use as an antifungal and anti-inflammatory agent for treatment of contact dermatitis, skin infections as well as its role as an anti-inflammatory agent. Thus, it was proposed that its beneficial effects and usage for the prevention of diseases related to fungal and inflammation need to be further examined [68].

In a screening of *C. lemon* and *C. sinensis* oils by disc diffusion method in combatting *M. furfur*, the diameter of the inhibition zone was found to be 50 and 20 mm, i.e., greater than those of reference antibiotics, gentamycin and streptomycin, at 16.5 and 17 mm, respectively. MIC of both lemon and orange oil against *M. furfur* was found to be 0.8 and 2.2 µL/mL. These findings support the use of *C. lemon* and *C. sinensis* oil as a traditional herbal medicine for the treatment of *P. versicolor* infection of the skin [82]. The evaluation of *Bursera morelensis* EO indicated inhibition of all the filamentous fungi. *F. monilifome* (IC50: 2.27 mg/mL) was the most sensitive fungal strain. This work provided scientific evidence for the antimicrobial activity of the *B. morelensis* [71].

In addition to in vitro approaches, several investigations were also conducted with in vivo models combined with in vitro approaches. Among the most widely covered EOs in this review for combined approaches were *P. jacquemontiana*, *C. martini*, *C. ambrosioides*, *C. citratus, T. ammi* and *A. houstonianum* Mill. *P. jacquemontiana* is an aromatic plant of the Hamamelidaceae family, and is well-recognized for its medicinal value, especially in terms of treating skin infections or eruptions, relieving body aches and minimizing dermatitis, along with its anti-inflammatory, antioxidant hepatoprotective activities, as shown in experimental rats [111,112,113]. The study revealed excellent antifungal characteristics of *P. jacquemontiana* EO against seven fungi and one yeast strain. The maximum inhibition zone was obtained with *Mucor piriformis* ATCC 52554 (19.83 ± 1.04 mm), while *F. solani* ATCC 36031 resulted in a minimum inhibition zone of 13.13 ± 1.03 mm and a maximum MIC value of 512 μg/mL [89].

The properties of this oil with a wide spectrum of antifungal capacity were believed to be strongly influenced by its chemical composition, i.e., mostly phenolic compounds. Past research has suggested that alcohols, phenolic compounds or aldehydes are capable of preventing fungi and yeast growth [114]. An in vivo study to evaluate the wound healing capacity of this EO was performed with Sprague-Dawley rats. Outstanding wound contraction rates were observed. This phenomenon was attributed to the presence of phytochemical compounds (phenolic compounds, flavonoids, terpenoids, alkaloids) which were reported previously to assist in wound contraction and facilitate epithelialization. Wound healing encompasses a critical step of increased proliferation and migration over the wound via re-epithelialization, that further strengthens this hypothesis [115]. Although *P. jacquemontiana* oil was presented as an effective approach for wound healing, this group of researchers also proposed that extended studies with regards to the mechanism involved should be undertaken in the future [89].

In another study, the capacity of *T. ammi* fruits EO was demonstrated for antidermatophytic activity using Swiss Albino mice model. *T. ammi* L. is a member of the Apiaceae family, that was known as an aromatic herb and spice grown in Egypt, Persia, Bangladesh, Afghanistan, Ethiopia and India [116]. These fruits possess numerous medicinal properties, including antimicrobial, anti-inflammatory, antioxidant, antiviral and anticandidal effects [117,118,119,120,121,122]. Chemical profiling of the oil revealed thymol as the major component, at 58.88%, followed by p-cymene (24.02%) and γ-terpinene (13.77%). The MIC value of *T. ammi* oil was noted in the range of 0.025–0.5 μL/mL against tested fungi. Maximum zone of inhibition was observed against *C. tropicum* (63.83 ± 0.166 mm) followed by *T. simii* (57 ± 0.288 mm), *T. rubrum* (51.33 ± 0.333 mm) and *C. indicum* (45 ± 0.577 mm). From a total of five concentrations tested, no irritations were observed at low concentrations, i.e., up to 3%, while three mice presented symptoms of mild erythema and all five mice exhibited well-defined erythema at 5% and 7% concentrations, respectively. This finding concluded that toxic side effects were not demonstrated at low concentrations. Based on the excellent antidermatophytic activity of the oil, *T. ammi* EO could be potentially used in the treatment of tinea or ringworm infections [90].

An EO originating from *A. houstonianum Mill* leaves was shown to be capable of inhibiting the growth of both *M. gypseum* and *T. mentagrophytes*, with MIC of 80 μg/mL. *A. houstonianum* is an annual or biannual herb and a member of the Asteraceae family, that has been utilized as a traditional medicine in many countries. The MIC value of *A. houstonianum* Mill EO when tested with multiple serial dilutions was 20 μg/mL for both (8:2) while 8 and 10 μg/mL value were obtained for *M. gypseum* and *T. mentagrophytes* (10:1), respectively. An in vivo study of the acute dermal toxicity effects with guinea pig indicated no changes in treated skin and fur, appearance along with the absence of diarrhea. In addition, dosage increase resulted in a reduction of noise sensitivity degree, pinch reaction, locomotion effects, as well as reactivity. However, a slight darkening of the liver, kidneys and heart was noticed with dosage increase. The observed effects indicated the suitability of *A. houstonianum* EO as an antidermatophytic compound. This supported a previous hypothesis that this EO may have depressant or sedative effects on the CNS at high doses [92,123].

*C. citratus* (DC.) *Stapf* (Poaceae family), commonly known as lemon grass, is a perennial tropical grass with thin, long leaves. It is one of the main medicinal and aromatic plants cultivated in Algeria, as well as in the tropical and sub-tropical regions of Asia, South America, and Africa [124]. *C. citratus* EO (LGEO) exhibits promising antifungal effect against *C. albicans*, *C. tropicalis* and *A. niger*, with different inhibition zone diameters (IZD), i.e., 35–90 mm, that increase proportionately with the increase in the oil volume. A chemical characterization of LGEO indicated the presence of geranial and neral as the major components, representing about 42.2 and 31.5%, respectively; its antifungal activity was associated with the presence of these compounds. The diameter of the growth inhibition zone (DGIZ) of LGEO in the vapor phase was found to be superior to the liquid phase, which is believed to be more advantageous in terms of application ease, elimination of the need for direct contact with the oil, and the requirement of using less oil. In vivo evaluation with Swiss albino mice showed considerable anti-inflammatory activity, where the degree of edema inhibition was similar for 10 and 100 mg/kg LGEO 90 min after oral administration (82.75 and 86.2%, respectively). This level of edema inhibition was comparable to that observed using 50 mg/kg oral doses of the standard reference drug, diclofenac (86.2%). The analysis of the inhibition degree of croton oil topical application resulted in noticeable edema on the left ear. On the whole, LGEO, as an antifungal and anti-inflammatory agent, is believed to be effective, both in the prevention and treatment of acute inflammatory skin conditions. In line with this potential, this research introduced ideas for additional evaluations of active constituents, specific action mechanisms and drug discovery for alternative antifungal therapies [91].

Commonly used antifungal methods include the disc diffusion technique, macro and microbroth dilution assays, the food poisoning method, the volatile release plate method, as well as the use of electron microscopy. Certain methods like disc diffusion, well diffusion and agar dilution methods have been shown to produce unreliable and inconsistent results [125,126]; this is primarily due to difficulties related to stable oil dispersion in aqueous media, lipophilic substance distribution in the aqueous medium, and various methods which are used to determine the number of viable microbes that will remain following the addition of EOs. In addition, true representation of activity is not possible with the disc and well diffusion methods, as different components diffuse through the agar at varying rates, and only the more water-soluble components, such as terpine 4-ol, are capable of agar penetration [125]. This phenomenon also occurs in the agar dilution method, where the incorporation of EO into agar can result in the loss of certain active compounds (e.g. terpinen-4-ol, linalool and limonene), either through evaporation or absorption into the petri dish plastic [127]. These techniques yield unreproducible outcomes for certain EOs, and are thus unreliable for assessments of the true behaviour of EOs. Usually, Tween 80 (a commercial polyoxyethylene sorbitan monooleate) will be added to emulate the oils for the study of the antifungal activity of hydrophobic and viscous EOs [128]. This may be linked to the fact that surfactants decrease the degradation rates of EOs, thereby causing the continued release of antimicrobial agents [129].

### 3.2. EOs in Nonconventional Approaches

Nonconventional technology is defined as a combinatorial approach of an EO with an absorption base (e.g., ointments). This approach was applied in one study that examined the effect of *L. multifora* (Lippia oil) in combination with six types of ointment base and incorporation of Tween 80 at four different concentrations to another set. The findings of this study revealed that the most effective inhibitory action was obtained when lippia oil (10% *w/w*) was combined with Hydrous Wool Fat Ointment BP (British Pharmacopoeia method) or Simple Ointment BP, with added Tween 80 (6% *w/w*) upon testing of *C. pseudotropicalis* and *C. albicans*. The incorporation of Tween 80 further enhanced the efficacy of this formulation, suggesting its suitability as a nonconventional treatment strategy [85].

The potential of *P. betle* EO in combination with polymer gel as a suitable dermatological formulation against multiple skin pathogens was studied. The reported results showed the highest sensitivity of *C. albicans* and *A. niger* toward *P. betle* EO, with inhibition up to 24 mm and 20 mm when tested at 20 µL concentrations, respectively. The inhibition zones were found to be less affected by gel incorporation, as confirmed by the antimicrobial results from the use of the final formulation [56]. Another potential treatment of candidiasis was tested on *B. tripartita*. An EO of *B. tripartita* (BTEO) was obtained via hydrodistillation of the aerial parts of the plant, which were then transformed into a gel formulation. The authors of this study concluded that both active phytoconstituents of *B. tripartita*, as well as the vehicle properties of the gel, contributed to the observed therapeutic activity. The highest inhibition was recorded on *C. tropicalis* and *C. krusei* with 3.7 mm difference compared to hydrogel without BTEO. Through the comparison, the hydrogel-based formulation combined with BTEO appeared to be the preferred mixture, exhibiting superior antifungal activity against all tested *Candida* strains and revealing the oil as a promising anticandidal agent [86].

Another study screened the potential of nine EOs, where *B. serrata* was cited for its superior activity, with maximal activity against *Trichophyton* spp. *B. serrata* EO was reported to contain mostly of *α*-thujene, *ρ*-cymene and sabinene compounds. The synergistic activity demonstrated with azoles in azole-resistant *C. albicans* indicated the suitability of this oil for use as an antifungal agent. However, as the study was limited in terms of its examination of the mechanism of action, individual constituent analyses and clinical studies, further investigations were deemed necessary [78]. Similarly, the antifungal activity of several EOs (black pepper, cardamom, cumin, Boswellia and Patcholi) were examined in another study against fluconazole-resistant fungi. The results revealed the effectiveness of Boswellia and Cardamom oil against *C. tropicalis* and *T. mentagrophytes*, respectively. Furthermore, a synergistic effects of a combination of Boswellia EO and fluconazole was noted, revealing itself to be the most effective of the applied approaches against *C. tropicalis*, even at 1:10 dilution and 100 μg/mL, respectively. As indicated by Sadhasivam and colleagues [78], an earlier study also proposed that the active compound, action mechanism, toxicity and stability be the focus of future investigations [88].

*R. officinalis*, *Lavandula x intermedia* “Sumian” and *O. vulgare subsp. hirtum* are among the most common medicinal plants in the Mediterranean region. They were widely used and have been synergistically tested for the treatment of candidiasis. Research on the HaCaT (normal cell line) and A431 (tumoral cell line) reported its safety for use as an antiproliferative agent. In vitro studies against *C. albicans*, *C. krusei* and *C. parapsilosis* showed an increase in the antifungal activity of clotrimazole-loaded nanoparticles (NLC). NLC containing Mediterranean EOs represents a promising strategy to improve drug effectiveness against topical candidiasis [87]. Infection of hair, globous skin and nails on humans is believed to be caused by keratinophilic fungi. Four types of an available commercial formulation called Itra-Bella (*Lonicera x bella zabel),* Kewda (*P. odoratissimus*), Rajnigandha (*P. tuberosa*) and Mogra (*J. sambac*) were tested against *A. flavus, T. mentagrophytes, T. tonsurans, T. verrucosum, E. floccosum* and *M. nanum*. The results showed better antifungal activity with EOs compared to synthetic drugs, i.e., terbinafine, itraconazole and fluconazole. *T. tosurans* was inhibited up to 47 mm by Itra-Bella, indicating that keratinophilic fungi could be successfully inhibited by Itra-Bella [62].

Several combination studies with in vitro and in vivo approaches also evaluated nonconventional technologies. One example is a study that investigated the potential of LGEO using nonconventional technology that incorporated the formulation of a topical hydrogel with lemongrass-loaded nano-sponges for antifungal action against *C. albicans* strain ATC 100231. Nano-sponges are a novel class of encapsulated nanoparticles that serves as an excellent delivery system in both the pharmaceutical and cosmeceutical industries [130]. This technology has received much attention in recent years, as it combines the benefits of both microsponges and nanosized-mediated delivery systems. Nanosponges with topical hydrogels have been shown to cause minimal irritation, fewer adverse effects as well as improved skin retention when compared with conventional topical delivery systems [57,131]. Similarly, this nonconventional approach was proven to be effective, i.e., the results confirmed the superior nonirritant and antifungal effects of the final selected hydrogel (F9) from among nine tested formulations. The MIC and MFC concentrations of LGEO using the broth macrodilution method was found to be 2 and 8 µL/mL, respectively. The results were promising, especially for the practical approach of using pharmaceutical formulations as a measure to minimize the risk of folk medicine usage in the crude form [93].

Another study by Gemeda and others conducted with EO from *C. martini* showcased broad-spectrum antimicrobial potency against all tested organisms, with MIC value ranging from 0.65 to 10 μg/mL. *C. martini* EO demonstrated absolute growth inhibitions of *Trichophyton mentagrophytes* and *T. rubrum* (>1% EO concentration), as well as *M. canis* and *T. verrucosum* (>4% EO concentration) [94]. This study incorporated a nonconventional approach, whereby various formulations were prepared and tested. A total of five base formulations (hydrophilic ointment, macrogol blend ointment, macrogol cream, simple ointment and white petrolatum base) were prepared with different concentrations of the EO by incorporating the oil into soft mass of different dermatological bases. The findings demonstrated that both hydrophilic and macrogol blend ointments with *C. martini* EO demonstrated superior activity upon tested organisms than relevant commercial drugs. Furthermore, a skin sensitization test with guinea pig showed no sign of irritation or sensitization when tested with 5% EO topical formulation. Hence, this product was believed to be a suitable alternative to established treatments, although extended studies are required to evaluate the safety, chronic toxicity and efficacy [94].

One more interesting study described the antifungal activity of *C. martini* and *C. ambrosioides* EO, along with their combinations, against several dermatophytes and filamentous fungi via in vitro as well as in vivo (guinea pig) tests. *C. martini* belongs to the Poaceae family, and is a known medicinal plant that possesses thermogenic, diuretic and insecticidal activity [132,133]. *C. ambrosioides* L. is a member of the Chenopodiaceae family and is widely used in Indian traditional medicines. Its antifungal and antiaflatoxigenic activities have been noted against certain storage fungi [51,132]. Its EO was prepared by mixing 1 mL of the oil with 100 g petroleum jelly. An in vivo study indicated significant efficacy of the EO against superficial mycosis, with a reduction in skin redness, lesion severity and dermatophyte occurrence upon ointment application, as confirmed via recurrence of hair growth at the infected site, compared to the control. The disease was successfully removed within 7–21 days via the application of EO ointments [95].

In addition, in vitro evaluation of both EO also inhibited the growth of *T. rubrum* and *M. gypseum*, with low MIC values that signified their potential role as alternatives to synthetic antifungal drugs. A chemical characterization indicated the presence of trans-geranoil (60.9%) and m-cymene (43.9%) as the major components of *C. martini* oil and *C. ambrosioides* EOs, respectively. One important element in this study was the more efficient synergistic effect obtained with a combination of both EOs in a 1:1 ratio than *C. ambrosioides* alone, indicating fungitoxicity alteration effects with the formation of combined chemical profiles. The MIC of EO and combination were shown to be around 150–500 ppm, which was comparatively lower than those of commercial drugs, which are usually higher, at around 1000–5500 ppm. In conclusion, this study highlighted the ability of both EOs, individually or in combination, to serve as a suitable strategy in the treatment of various superficial mycoses in humans, and recommended that clinical trials be performed [95].

### 3.3. EOs in Clinical Interventions

The effectiveness of EOs for the treatment of skin fungus has also been investigated through clinical interventions, in addition to in vitro and in vivo studies. In our review, we came across two interesting clinical research projects that had evaluated the antifungal activity of EOs originating from distinct plant species. In the first intervention, the antifungal activity of *M. communis* EO against *Malassezia* sp. isolated from the skin of patients with *Pityriasis versicolor* was examined. *Malassezia* sp. is a class of lipophilic yeasts that covers part of the skin microflora in humans and other warm-blooded animals [134]. However, the presence of this yeast is also known to lead to diseases such as inflammation, systemic infections or uninflamed lesions. Among these, uninflamed lesions in the presence of excess fungal load represent a disease known as *Pityriasis versicolor* (PV) [134,135]. PV is sometimes also known as tinea versicolor, and is characterized by the development of hypo or hyperpigmented scaly spots [134,136].

*Malassezia* furfur is recognized as one of the main causative agents of PV, from among the 14 recognized *Malassezia* species [134,135]. The EO applied in this study was isolated from *Myrtus communis* (Myrtaceae), which has been widely used in various medicinal remedies. To date, this myrtle EO is known to contain 1, 8-cineole and α-pinene. Its antifungal activity against several species (e.g. *Rhizoctonia solani, F. solani, A. flavus*, *Colletotrichum lindemuthianum*, *F. culmorum*, *C. albicans*) [137,138,139,140] has been documented. In this prospective case-series study, clinical isolates of PV were collected from 41 patients that had been deprived of treatment for at least two weeks. The clinical samples revealed the presence of 86 yeast colonies with seven identified species: *M. furfur* (42.5%), *M. sympodialis* (23.5%), *M. sllooffiae* (13.9%), *M. globose* (7.5%), *M. obtusa* (6%), *M. japonica* (4%) and *M. restricta* (2.5%). The antifungal inhibition revealed excellent activity of *M. communis* against *Malassezia* sp., especially *M. furfur* and *M. sympodialis*, at 96% and 83%, respectively. The findings of this study suggest the potential of *M. communis* EO as an alternative to conventional antifungal drugs, especially in the treatment of PV. Such an approach was considered cheaper, safe and nonhepatotoxic or nonnephrotoxic. Nevertheless, this study also addressed its limitations, and proposed the need for in vivo studies to confirm the effectiveness of this EO [97].

The second investigation focused on the antifungal activity of EO obtained from *C. longa* L. waste leaves. *Curcuma longa* L. is a member of the Zingiberaceae family that grows mostly in Sri Lanka, Indonesia and many parts of India. *Curcuma longa* L. is usually called turmeric, haldi or Indian saffron, with well-known usage as a spice, as well as a treatment against common skin diseases [141]. In a study by Pandey and colleagues, *Curcuma longa* L. EO exhibited positive effects against several human pathogenic fungi (Pandey et al. 2010). The selected patients in this study had been diagnosed with *Tinea corporis*. *Tinea corporis* is classified as a form of dermatophytic infection of glabrous skin. *Trichophyton* and *Microsporum* species were identified as the cause of this infection. *Tinea corporis* infection generally occurs within the stratum corneum of the epidermis, and may be transmitted via direct contact with infected individuals or animals [96,142,143].

Chemical characterization of the oil indicated the presence of terpinolene, α-phellendren, terpinen-4-ol and sabinyl acetate as the main components. It was also noted that the oil destroyed both *M. gypseum* and *T. mentagrophytes* within 5 s. In addition, antifungal activity of the oil was observed with other species such as *E. floccosum, M. nanum, T. rubrum* and *T. violaceum.* This study highlighted the significant potential of *C. longa* leaf EO based on in vitro and in vivo assays; additionally, strong fungicidal action, extended shelf-life, tolerability of higher inoculum density, thermal stability and a wider range of antidermatophytic activity without any adverse effects were noted. Intervention studies that involved the topical application of the oil formulation gave positive insights into its efficacy; complete healing was observed in 72% of cases after three weeks of treatment, without any relapse when patients were examined two months later. Hence, the study proposed the application of *C. longa* leaf EO as an inexpensive and effective formulation for commercial use [96].

### 3.4. EOs in Other Nonconventional Approaches and Their Advantages

The development of numerous nonconventional approaches in antifungal therapies indicates great potential. In addition to the reviewed articles, one prominent technique that has gained attention is the application of cyclodextrins via encapsulation. This technique involves the formation of inclusion complexes of selected oils with cyclodextrins (CDs). In a study performed by Torres-Alvarez and colleagues, concentrated orange oils were subjected to encapsulation with β-CD and evaluated for their antifungal activity. This study noted that the highest antifungal activity inhibition was achieved at 10 mg/mL, respective to 10× fraction orange oil in ratios of 12:88 and 16:84 (COEO: β-CD). Hence, the encapsulation technique proved to be beneficial toward inhibition, although the researchers proposed further studies to confirm the formation of the complex and the encapsulated compounds [144].

Pickering emulsions (PEs) are widely sought after at present, compared to conventional emulsions. PEs comprise particle-stabilized emulsions in which solid particles tend to adsorb instantly on the oil-water interface, resulting in a shell-like structure on the PE droplet surface. There are various advantages associated with the PE approach, e.g., improved stability in relation to the higher adsorption energy of solid particles, as well as appropriate emulsion droplet size, that could assist in longer retention of the antifungal drug at the treatment site. The approach of PE was demonstrated in one study that evaluated its effectiveness as an alternative therapy for onychomycosis topical treatment, also known as fungal nail infections. For this purpose, a nanotechnological approach was applied to formulate PEs in combination with azole derivative (tioconazole) and *Melaleuca alternifolia* EO tested against *C. albicans* and *T. rubrum*. The findings showed that PEs are an excellent choice for onychomycosis topical treatment compared to conventional emulsions or ethanolic solutions of reference drugs [145].

Nonconventional techniques are have gained broader recognition in recent years due to their excellent benefits. In this review, we have discussed various nonconventional approaches, and noted their superiority compared to the traditional methods. In general, we would like to highlight the fact that strategies involving nonconventional treatments are useful to reduce risks associated with the crude forms of folk medicines, along with lesser adverse effects and higher skin retention for better targeted drug delivery.

## 4. Materials and Methods

### Search Strategy

Original articles were identified through searches of three databases (PubMed, WoS and Google Scholar) for the period of 2010 to 2020 using (“skin”[MeSH Terms] OR “skin”[All Fields]) AND (“antifungal agents”[Pharmacological Action] OR “antifungal agents”[MeSH Terms] OR (“antifungal”[All Fields] AND “agents”[All Fields]) OR “antifungal agents”[All Fields] OR “antifungal”[All Fields] OR “antifungals”[All Fields] OR “antifungic”[All Fields] OR “antifungical”[All Fields]) AND (“oils, volatile”[MeSH Terms] OR (“oils”[All Fields] AND “volatile”[All Fields]) OR “volatile oils”[All Fields] OR (“essential”[All Fields] AND “oil”[All Fields]) OR “essential oil”[All Fields]). Publications with abstracts were reviewed; the search was limited to studies published in the English and Malay languages. Papers on in vitro, in vivo, human and animal studies, and related to plant-based antifungal medication were included. Review articles and letters to the editor were excluded. Duplicate articles were eliminated.

## 5. Conclusions

In conclusion, our review underlines the fact that the importance of antifungal agents EOs is widely recognized. Their roles in reducing the severity of fungal infections vary according to species and origin. Based on our review, we strongly believe that EOs should be explored for commercial applications as alternatives to over-the-counter antifungal agents. In addition, commercial applications could be further enhanced with nonconventional strategies in combination with other components, such as fluconazole and Tween 80. Hence, it is vital that efforts continue for the development of EO-based skin antifungal therapies.

## Figures and Tables

**Figure 1 molecules-26-01093-f001:**
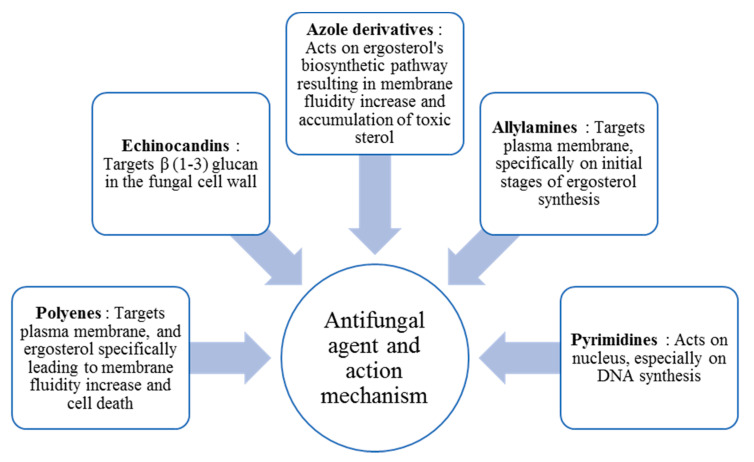
Conventional antifungal agents and their mechanisms of action (Adapted from [19]).

**Figure 2 molecules-26-01093-f002:**
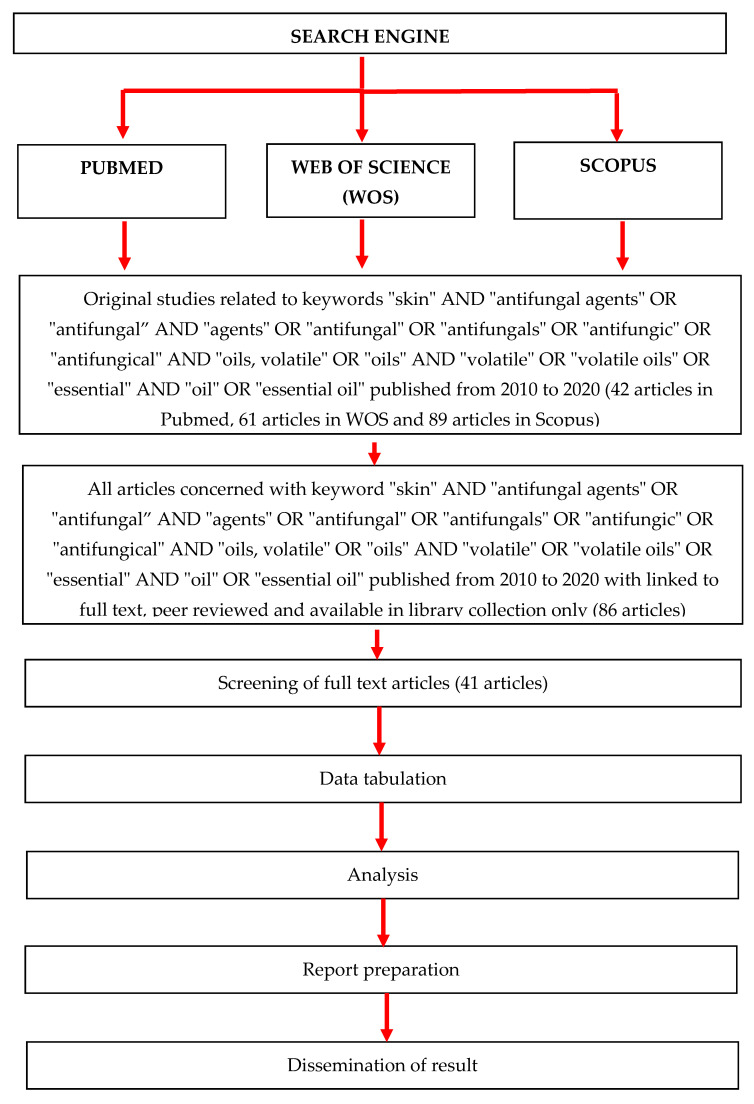
PRISMA flow diagram of the identification of literature for inclusion in this review.

**Table 1 molecules-26-01093-t001:** Potential of EOs as antifungal agents in human skin disease.

No.	Techniques	Reference	Type of Skin Infections/ Fungi Species	Essential Oils	Methods	Findings	Conclusion
1.	In vitro—Conventional technology	[58]	Skin infections (*C. albicans*, *C. parapsilosis*)	Ten EOs: oregano, thyme, clove, arborvitae, cassia, lemongrass, melaleuca, eucalyptus, lavender, and clary sage	EO compositions were determined using gas chromatography-mass spectrometry (GC/MS). The genotoxic effects on healthy human keratinocytes HaCaT of the tested EOs were evaluated using the comet assay	Oregano (76.73% carvacrol), thyme (53.26% thymol), cassia (77.83% eugenol), lemongrass (45.72% geranial) and arborvitae (55.96% methyl thujate) showed very strong antifungal activity against all tested strains. The results revealed that none of the EO induced significant DNA damage in vitro after 24 h.	These results showed that these ten EOs may be effective in preventing the growth of the drug-resistant microorganisms which are responsible for wound infections. This EO could be used as a potential source of safe and potent natural antimicrobial and antioxidant agents in the pharmaceutical and food industries.
2.		[59]	Dermatophytosis by dermatophytes (*Trichophyton rubrum* PTCC (Persian Type Culture Collection)5143, *T. mentagrophytes* PTCC5054, *Microsporum* canis PTCC5069, *M. gypseum* PTCC 5070, *T. schoenleinii* PTCC5221, *T. verrucosum* var. album PTCC 5056)	*Artemisia sieberi* oils	Analysis of five *A. sieberi* oils (different harvesting times and distinctive collecting locations) by gas chromatography-flame ionization detector (GC-FID) and GC-MS. Antifungal activities of *A. sieberi* oils were evaluated against different dermatophytes. The oils were examined for antidermatophyte activity by the poisoned food technique. MIC and MFC were determined by broth microdilution assay.	The results showed that the harvesting time and collecting location affect the chemical composition and oil yields. The main components were santolina alcohol, camphor, α-thujone and β-thujone. No significant difference between the desirable antidermatophyte activities of *A*. *sieberi* oils with different chemical compositions. *M*. *gypseum*, *T*. *rubrum* and *M*. *canis* had more sensitivity than others to *A*. *sieberi* oils.	*A*. *sieberi* can be used as topical antifungal agent for the treatment of skin dermatophyte infections.
3.	In vitro—Conventional technology	[60]	Various skin infection due to pathogens (*C. albicans* ATCC 10231, and dermatophyte *T. rubrum* SNB-TR)	*Coridothymus capitatus* L.,	EOs composition analysis by GC/MS. Antimicrobial activity was measured using a broth microdilution method.	EO with high amounts of thymol and carvacrol possessed the strongest antimicrobial activity. These two compounds demonstrated interesting antifungal efficacy against the filamentous fungus *T. rubrum*.	This study highlights the in vitro antimicrobial activity of some Lebanese Lamiaceae EOs against human pathogens. The antimicrobial potential originates from their high content of either thymol or carvacrol. The results confirmed that some of the Lamiaceae species used in Lebanon ethnopharmacological practices as antimicrobial agents possess antibacterial and antifungal potential.
				*Lavandula stoechas* L.,			
				*Lavandula angustifolia* Mill.,			
				*Mentha spicata* L. subsp. condensata,			
				*Origanum syriacum* L.,			
				*Rosmarinus officinalis*,			
				*Salvia fruticosa* Miller.,			
				*Satureja cuneifolia* Ten.,			
				*Satureja thymbra* L.,			
				*Thymbra spicata* L., and *Vitex agnus-castus* L.			
4.		[61]	Skin disease due to six fungal strains (*C. tropicalis, C. albicans, Rhizomucor miehei, C. glabrata, A. niger* and *A. fumigates*)	*Myristica fragrans*	Volatile oil from the leaves was isolated and characterized by GC-MS. Oil extracts were subjected to antimicrobial assay followed by the Kirby Bauer method.	The most abundant constituents identified were β-pinene, α-pinene and α-thujene. The oil showed significant inhibitory activity against the fungus *C. tropicalis* (1.3 cm), *C. albicans* (0.8 cm), *R. miehei* (0.6 cm) and *C. glabrata* (0.6 cm). No inhibitory activity was detected against *A. niger* and *A. fumigates.*	The oil showed significant activity against the tested fungi that suggest the potential of *Myristica fragrans* EO against these diseases.
5.		[62]	Keratophilic fungi, a type of dermatophytes causing infection to hair, glabrous skin and nails (*A. flavus*, *T. mentagrophytes, T. tonsurans, T. verrucosum, Epidermatophyton floccosum* and *Microsporum nanum)*	Four different commercially available *Itra* (Volatile plant oils): *Bella (Lonicera x bella zabel), Kewda (Pandanus odoratissimus), Rajnigandha (Polianthes tuberosa) and Mogra (Jasminum sambac)*	The strains were identified by a morphological study of culture and microscopic analyses of developed colonies. Antifungal activity of *Itra* was tested against isolated fungi using agar-well diffusion method.	Better antifungal activity than existing antifungal drugs like terbinafine, itraconazole and fluconazole (taken as control). The maximum effect was shown by Bella against *T. tosurans* with inhibitory zone of 47 mm.	The paper concluded that *Itra* can be used as an antifungal agent against prevalent keratinophilic fungi.
6.	In vitro—Conventional technology	[63]	Various skin diseases (*C. krusei* H9, *C. guillermondii* MAT23, *C. albicans* ATCC 10231, *C. tropicalis* ATCC 13803, *C. parapsilosis* ATCC 90018, *Cryptococcus neoformans* CECT 1078, *T. mentagrophytes* FF7, *Microsporum canis* FF, *T. rubrum* CECT 2794, *M. gypseum* CECT 2908, *A. niger* F01, *A. fumigatus* F05, *A. fumigatus* F07, *A. fumigatus* F17, *A. flavus* F44, *A. niger* ATCC 16404, *A. fumigatus* ATCC 46645	Four type oils obtained from *Thymus zygis* subsp. *Sylvestris* (collected at four sites)	Chemical profiling using GC-FID and GC-MS. A macrodilution broth method was used to determine the MIC and minimum lethal concentrations (MLC) for yeasts and filamentous fungi. Antifungal activity of four oils (samples 1–4) was evaluated against yeasts, *Aspergillus* and dermatophyte strains. Cell viability by MTT assay.	Four chemotypes were characterized: carvacrol, thymol, geranyl acetate/geraniol and linalool. The results demonstrated that the oil with high amounts of carvacrol, which have stronger antifungal activity against dermatophyte strains, showed no cytotoxic effect, at concentrations ranging from 0.08 to 0.16 μL/mL, for as long as 24 h.	The potent antifungal activity of *T. zygis* subsp. *sylvestris* oil against dermatophyte strains was demonstrated, which justifies the widespread use of this plant in traditional medicine. However, further investigation is needed to evaluate the suitability of its antifungal properties in practical applications, namely, in fungal diseases involving mucosal and cutaneous infections, and also as an alternative to synthetic fungicides.
7.		[64]	Dermatophytosis or ring worm infections (*T. rubrum* PTCC 5143, *T. mentagrophytes* PTCC 5054, *M. canis* PTCC 5069, *M. gypseum* PTCC 5070, *T. schoenleinii* PTCC 5221, *T verrucosum* var. *album* PTCC 5056)	*Satureja khuzistanica*	Chemical profiling using GC-MS. Antifungal activities of extracts and EO were screened by disc diffusion method, microbroth dilution assay and food poisoning method. MIC and MFC of extracts and EO were determined by broth microdilution assay.	Carvacrol (94.5%) was the main component of *S. khuzistanica* EO. The antifungal activity of *S. khuzistanica* EO (MIC: 40–190 µg/mL) was higher than its ethanol extract (MIC: 40–770 µg/mL); while *S. khuzistanica* aqueous extract had no antifungal activity against dermatophytes (MIC = 1550–3100 µg/mL).	The oil can be used as a topical treatment for the control or treatment of skin fungal infections. *S. khuzistanica* EO or its ethanol extract can be further extended to more deeply explore their antifungal activities.
8.	In vitro—Conventional technology	[65]	Management of dermatophytosis (*Candida krusei* H9, *C. guillermondii* MAT 23, *C. albicans* ATCC 10231, *C. tropicalis* ATCC 13803, *C. parapsilosis* ATCC 90018, *C. neoformans* CECT 1078; *T. mentagrophytes* FF7, *M. canis* FF, *E.* FF9, *T. rubrum* CECT 2794, *M. gypseum* CECT 2908, *A. flavus* F44, *A. niger* ATCC 16404, *A. fumigatus* ATCC 46645	*D. carota* subsp. *gummifer* (Syme) Hook.f.	Chemical profiling (GC-MS). MICs and MLCs for yeasts and filamentous fungi were determined using macrodilution broth assays. Anti-inflammatory and nitric oxide scavenging activity. Toxicity was investigated using MTT assay in several mammalian cell lines: macrophages (Raw 264.7), keratinocytes (HaCat), hepatocytes (HepG2) and microglia (N9).	The oil was characterized by high contents of monoterpenes (83.9%), the major compounds being geranyl acetate (37.0%) and α-pinene (30.9%). The daucane sesquiterpene, carotol, was also found in relatively high amounts (11.0%). The oil was particularly active against dermatophytes and *C. neoformans*, with MIC values ranging from 0.32 to 0.64 μL/mL. The oil demonstrated a strong anti-inflammatory activity by inhibiting nitric oxide (NO) production in both lipopolysaccharide (LPS)-triggered macrophages and microglia cells. triggered macrophages and microglia cells. We detected a cytotoxic effect only for the highest concentrations of the oil, thus ensuring a safe toxicological profile at bioactive concentrations.	These results suggest that *D. carota* subsp. *gummifer* EO should be explored as a natural source of antifungal and anti-inflammatory drugs with potential application both at the peripheral and central nervous system levels, thus supporting in vivo studies focused in the management of dermatophytosis and/or inflammatory-related diseases.
9.	In vitro—Conventional technology	[66]	Various skin diseases *(C. albicans* ATCC 10231, *C. parapsilosis* ATCC 90018, *C. tropicalis* ATCC 13803, *C. neoformans* CECT 1078, *C. guillermondii* MAT2, *C. krusei* H9, *A. niger* ATCC 16404, *A. fumigatus* ATCC 46645, *A. flavus* F44, *E. floccosum* FF9*, T. mentagrophytes* FF7, *M. canis* FF1, *T. rubrum* CECT 2794, *M. gypseum* CECT 2908, *T. mentagrophytes* var. *interdigitale* CECT 2958, and *T. verrucosum* CECT 2992)	*Salvia officinalis* L.	Chemical profiling by GC and GC-MS. Antifungal activity was evaluated against yeasts, dermatophyte and *Aspergillus* strains by broth macrodilution methods. Cell viability by MTT assay. Anti-inflammatory potential was evaluated by measuring NO production using LPS-stimulated mouse macrophages.	The main compounds of *S. officinalis* oils were 1,8-cineole and camphor. The dermatophyte strains showed more sensitivity to these oils when compared with *Candida* and *Aspergillus* strains, particularly for *T. rubrum* and *E. floccosum* with MIC of 0.64 μL/mL. Among the tested yeasts, *Cryptococcus neoformans* showed more sensitivity with MIC of 0.64 μL/mL. The oils revealed antifungal activity against dermatophyte strains and significantly inhibited NO production stimulated by LPS in macrophages, without affecting cell viability, in concentrations up to 0.64 μL/mL.	This is the first report addressing the in vitro anti-inflammatory potential of *S. officinalis* oil. These findings demonstrated that bioactive concentrations of *S. officinalis* oils do not affect mammalian macrophages or keratinocytes viability, making them suitable for use in skin care formulations for cosmetic and pharmaceutical purposes.
10.		[67]	Various skin diseases (*C. albicans ATCC 10231, T. rubrum SNB-TR1, T. mentagrophytes SNBTM1, T. soudanense SNB-TS1, T. violaceum SNB-TV1* and *T. tonsurans SNB-TT1)*	*Abies cilicica, Cupressus sempervirens, Juniperus excelsa, Juniperus oxycedrus, Cedrus libani* and *Cupressus macrocarpa* gold crest	Chemical profiling by GC and GC-MS. The MICs of these EOs were determined using the broth microdilution technique.	The EOs showed the most interesting bioactivity on the dermatophytes species (MIC values 32–64 µg/mL). Each of the major compounds of *C. macrocarpa*, as well as artificially reconstructed EOs, were tested on *T. rubrum*, showing a contribution of the minor components to the overall activity. The main components were α-pinene and sabinene.	Lebanese conifer EOs showed interesting antimicrobial activity, particularly against *Trichophyton* species. These EOs should therefore be considered for a possible integration in antimicrobial preparations intended to be used in the treatment of superficial infections.
11.	In vitro—Conventional technology	[68]	Contact dermatitis and skin infections (*C. krusei* H9, *C. guillermondii* MAT23, *C. albicans* ATCC 10231, *C. tropicalis* ATCC 13803, *C. parapsilosis* ATCC 90018, *C. neoformans* CECT 1078, *E. floccosum* FF9, *T. mentagrophytes* FF7, *M. canis* FF1, *T. rubrum* CECT 2794, *Microsporum gypseum* CECT 2908, *A. flavus* F44, *A. niger* ATCC 16404, *A. fumigatus* ATCC 46645)	*Distichoselinum tenuifolium*	Chemical profiling by GC and GC-MS. Antifungal activity MIC and MLC were evaluated against yeasts, dermatophyte and *Aspergillus* strains. Cell viability by MTT assay. Anti-inflammatory potential was evaluated by measuring NO production induced by LPS, in the absence or in the presence of the oil, in a mouse macrophage cell line.	The oils are predominantly composed of monoterpene hydrocarbons, with myrcene being the main compound (47.7–84.6%). The oils revealed significant antifungal activity against *C. neoformans* and dermatophyte strains, and significantly inhibited NO production stimulated by LPS in macrophages without affecting cell viability at concentrations ranging from 0.64 L/mL to 1.25 L/mL.	These findings add significant information to the pharmacological activity of *D. tenuifolium* EO, specifically, to its antifungal and anti-inflammatory properties, thus justifying and reinforcing the use of this plant in traditional medicine, mainly for the treatment of contact dermatitis, characterized by a strong inflammatory component, and skin infections. Therefore, their beneficial effects and use in disease prevention, especially when related to fungal infections and inflammation, should be explored in more depth.
12.	In vitro—Conventional technology	[69]	Skin infections (*T. rubrum* and *T. mentagrophytes*)	*Agathosma betulina* and *Coleonema album*	EOs analysis by GC-MS. Antifungal capacity was evaluated by volatile release plate method. The mode of action of the EOs volatiles was elucidated using SEM.	The major components of the EO of *A. betulina* were limonene (29.8%), menthone (21.6%), and isomenthone (14.7%), while of *C. album* were pinene (27.4%), and myrcene (14.5%). There was inhibition/reduction of fungal growth in all plates exposed to EO volatiles. However, the rates of inhibition varied among the fungal species, EO and different volumes tested. The volatiles from *A. betulina* EO showed a remarkable inhibitory effect, with the highest inhibition recorded at 40 µL. The best inhibition was recorded in *T. rubrum* exposed to volatiles at 40 µL of *A. betulina* EO with a fungal growth index of 2.3%, indicating its strong fungicidal effects. *A. betulina* EO (40 µL) resulted in the destruction of hyphae and spores of *T. rubrum*.	The EO volatiles of *A. betulina* had good inhibitory effects on the mycelia growth of *T. rubrum* and *T. mentagrophytes*. The fungicidal effect of EO volatiles on *T. rubrum* was particularly noteworthy. The mode of action of *A. betulina* EO volatiles on the growth of *T. rubrum* mycelia was shown to be through the inhibition of spore growth and production and alteration of the morphology by destruction of the hyphae and spores. EO induced different actions on the mycelia at different concentrations. Limonene and menthone, as the major components of *A. betulina* EO, are probably responsible for this activity, as they have been previously shown to possess antifungal activity. The remarkable activity recorded in this study gives an insight into the potential of *A. betulina* EO as a skin antifungal.
13.		[70]	Skin infections (*T. rubrum* and *T. mentagrophytes*)	*C. album* and *C. pulchellum*	EOs analysis by GC-MS. Antifungal capacity of EOs was evaluated by volatile release plate method.	Terpenes formed the major components of the EOs. Caryophyllene (24.91%), *trans*-β-ocimene (9.49%), β-myrcene (7.96%) and octahydro-7-methyl-3-methylene-4-(1-methylethyl)-β-copaene (7.53%) are the major components of *C. album* EO while bicyclo[3.1.0]hex-2-ene,4-methyl-1-(1-methyethyl)-β-phellandrene (32.04%), (+)-3-carene (10.88%), β-ocimene (9.46%), γ-elemene (8.95%) and α-pinene (7.43%) are the major components of *C. pulchellum* EO. The EO from both plants inhibited the growth of *T. rubrum* in vitro with final mycelia diameter of 0.3 cm and fungal growth index (FGI) of 0%.	This study revealed the therapeutic value of *C. pulchellum*. *Coleonema album* and *C. pulchellum* should be considered as potential plants for use in skin ointment. However, toxicity studies of *C. pulchellum* EO need to be carried out in order to ensure their safety for topical application.
14.	In vitro—Conventional technology	[71]	Skin infections (*Fusarium sporotrichioides*, *F. moniliforme, T.n mentagrophytes, A. niger*, and *Rhizoctonia lilacina*)	*Bursera morelensis*	EO analysis by GC-MS. Antifungal activity.	GC-MS demonstrated the presence of 28 compounds. The five principal compounds were α-Fellandrene (32.69%), β-Fellandrene (14.79%), o-Cymene (8.71%), Isocaryophyllene (7.48%) and α-Pinene (5.82%). The EO inhibited all the filamentous fungi except *F. sporotrichioides*. *F. monilifome* (IC50 = 2.27 mg/mL) was the most sensitive fungal strain.	This work provided evidence of the antimicrobial activity of the *B. morelensis* EO, providing further scientific support for the traditional use of this species.
15.		[72]	Skin infections (*C. albicans, Candida stellatoidea* and *Candida torulopsis*)	*Physalis angulata*	Antifungal activity by agar well diffusion method.	*C. torulopsis, C. stellatoidea* and *C. albicans* were susceptible to the EO from the aerial and root part of the plant (MIC: 3.75–4.0 mg/mL).	The observed inhibition of selected fungi by oils of *P. angulata* makes it a promising antimicrobial agent.
16.	In vitro—Conventional technology	[73]	Dermatophytes and fungi infections (*Microsporum gypseum, M. canis, Arthroderma cajetani, T. mentagrophytes, T. tonsurans, E. floccosum, Botrytis cinerea, Pythium ultimum*)	*Marrubium vulgare* L.	EO analysis by GC/MS. Antifungal activity by transplanting mycelium disks from a single culture in stationary phase.	EO was high in eugenol (15.29%). The antifungal activity of different extracts, marrubiin and EO at two concentrations (20 and 100 μg/mL) showed about 50% inhibition for *T. mentagrophytes* and *E. floccosum* for marrubiin at 100 μg/mL. The antiphytopathogenic activity was also evaluated; only marrubiin had activity against *B. cinerea* at the highest dose (32.40%).	*M. vulgare* and marrubiin can be used as natural antioxidants and antifungal agents for the treatment of skin dermatophyte infections. Complementary investigations should be conducted to assess the effectiveness of this compound.
17.		[74]	Skin infections (*C.a albicans 1, C. albicans 2, T. rubrum 1, T. rubrum 2, T. violaceum*, and *T. soudanensis*)	*Eugenia caryophylla* and *Mentha sp cf piperita*	EOs analysis by GC-MS. Antifungal activity by broth microdilution method M27-A3 and M38-A.	*E. caryophylla* was mainly composed of eugenol (80.0%), square-caryophyllene (8.3%), and eugenol acetate (6.7%) while *M. sp cf piperita* was characterized by piperitone (67.5%), menthol (10.0%) and beta-phellandrene (5.8%). *E. caryophylla* EO exhibited the highest antifungal activity with MIC and MFC of 0.25 μL/mL and 0.125 μL/mL for filamentous fungi and MIC of 0.5 μL/mL for both yeast strains while MFC value was 1 μL/mL for one yeast strain and not determined for the second. MFC of *M. sp cf piperita* EO showed weak activity with a MIC of 2.5 μL/mL on *Tricophyton* strains, while no activity was exhibited on *C. albicans* strains.	The results of this work can be used to confirm their traditional uses and can also be proposed as natural ingredients to some industries to treat superficial infections.
18.	In vitro—Conventional technology	[75]	Skin diseases (*E. floccosum, M. canis, M. gypseum, T. mentagrophytes, T. mentagrophytes var. interdigitale, T. rubrum, T. verrucosum*, *C. albicans*, C. *guilliermondii, C. krusei, C. parapsilosis, C. tropicalis, C. neoformans, A. flavus, A. fumigatus, A. niger*).	*Juniperus communis* subsp. *alpina* (Suter) Celak	EO analysis by GC-MS. Antifungal activity by broth macrodilution method. Cytotoxicity was tested in a human keratinocyte cell line (HaCaT) through MTT assay.	*J. communis* subsp. *alpina* needles EO was predominantly composed of monoterpene hydrocarbons (78.4%), with the main compounds being sabinene (26.2%), α-pinene (129%) and limonene (10.4%). Antifungal activity demonstrated the potential of needle oil against dermatophytes, particularly for *M. canis* and *T. rubrum*, with MIC and MLC of 0.32 μL/mL. Evaluation of cell viability showed no significant cytotoxicity in HaCaT keratinocytes at concentrations between 0.32 and 0.64 μL/mL.	These results show that it is possible to find appropriate doses of *J. communis* subsp. *alpina* oil with both antifungal activity and a very low detrimental effect on keratinocytes.
19.		[76]	Skin infection sporotrichosis (*Sporothrix schenckii fungus*)	Turmeric (*Curcuma longa*)	EO analysis by GC-MS. Antifungal activity by disc diffusion method.	GC MS analysis revealed the presence of more than 20 component, including Ar-turmerone and dehydro Ar-turmerone. Activity against *S. schenckii* fungus showed zone of growth inhibition was 9 mm by the sample oil after 8 days and 11 mm after 10 days.	No conclusion was presented.
20.		[77]	Skin infections (*T. rubrum* and *T. mentagrophytes)*	*Lavandula luisieri* and *Cymbopogon citratus*	EOs analysis by GC-MS. Antifungal activity. Inhibition of conidial germination and antifungal drug/EOs combination assay.	EOs were characterized by a high amount of oxygenated monoterpenes in their composition. Strong antifungal activity was observed for the majority of clinical strains, and fungicidal activity was demonstrated. Positive interaction between *L. luisieri* EO combined with terbinafine was observed against terbinafine-resistant strain (Tr ATCC MYA-4438). Significant reduction of the germination was observed above 100 μg/mL. Both oils were safe to macrophage mammalian cells at tested concentration.	This study describes the antifungal activity of *L. luisieri* and *C. citratus* EOs against dermatophytes, which could be useful in the design of new formulations for topical treatments. This study also proposes *L. luisieri* as a suitable oil for combined treatment against terbinafine-resistant fungal strains, although further in vivo experiments are needed to evaluate the therapeutic efficacy and safety of the EOs in combination with antifungal drugs.
21.	In vitro—Conventional technology	[78]	Skin diseases (*Propionibacterium acnes*, *Malassezia* spp., *C. albicans* and *Trichophyton* spp.)	Nine EOs (*Cyperus scariosus* R. Br., *Syzgium aromaticum* L. Merr et L.M. Perry*, Carum carvi* L., *Coriandrum sativum* L.*, Boswellia serrata* Roxb. Ex Colebr., *Syzgium cumini* L. Skeels, *Elettaria cardamom* L. Maton, *Occimum sanctum* L. and *Piper nigrum* L.)	EOs analysis by GC-MS. Antifungal activity by MIC (microbroth dilution) and zone of inhibition (ZOI) test. Synergistic activity by Etest and ZOI.	*B. serrata* EO indicates a major composition of α-thujene, ρ-cymene and sabinene. *B. serrata* EO demonstrated superior antimicrobial activity against all the micro-organisms, and surprisingly, it showed maximum activity against *Trichophyton* spp. Additionally, *B. serrata* EO combination with azoles showed synergistic activity against azole-resistant strain of *C. albicans*. These broad-spectrum antimicrobial activities of *B. serrata* EO will make it an ideal candidate for topical use.	*B. serrata* EO is a plausible therapeutic agent for the management of skin, scalp and nail infections. Further studies are needed to determine the mode of action and its safety for clinical use, and an analysis should be undertaken of its individual constituents.
22.	In vitro—Conventional technology	[79]	Skin infections by yeast (*C. albicans* 31, *C. tropicalis* 32, *C. glabrata* 33, *C. glabrata* 35 *and C. glabrata* 38)	Six commercial lemon EOs (ETJA, Vera-Nord, Avicenna-Oil, Dufti by Gies, Art, and Croce Azzurra	EOs analysis by GC-MS. Antifungal activity by MIC. GC-MS analysis.	Various lemon EOs had different chemical compositions, but mostly, they contained almost exclusively terpenes and oxygenated terpenes. The tests show that the antifungal potential of lemon EOs against *Candida* yeast strains is related to the high content of monoterpenoids and the type of strains. From six tested commercial oils, only four (ETJA, Vera-Nord, Avicenna-Oil and Aromatic Art) showed antifungal potential against three *Candida* species (*C. albicans, C. tropicalis* and *C. glabrata*). Vera-Nord and Avicenna-Oil showed the best activity, and effectively inhibited the growth of the *C. albicans* strain across the full range of concentrations used.	This study characterized lemon EOs, which could be used as natural remedies against candidiasis caused by *C. albicans*.
23.		[80]	Skin diseases (*M. canis* KCTC 6348, 6349, 6591, *T. rubrum* KCTC 6345, 6352, 6375, *T. mentagrophytes* KCTC 6077 and 6085)	*Lonicera japonica* Thunb.	Antidermatophytic activity of EO by poisoned food technique. MIC by agar dilution method. Spore germination and growth kinetics assay.	*L. japonica* EO (1000 ppm) revealed 55.1–70.3% antidermatophytic effect against *Microsporum canis* KCTC 6348, 6349, 6591, *T. rubrum* KCTC 6345, 6352, 6375, *T. mentagrophytes* KCTC 6077 and 6085, respectively, with MIC ranging from 62.5–500 μg/mL. The oil had strong detrimental effect on the spore germination of all the tested dermatophytes, as well as concentration and time-dependent kinetic inhibition of *M. canis* KCTC 6348.	The results demonstrated that *L. japonica* oil could be a potential source of natural fungicides to protect human and animals from fungal infections.
24.	In vitro—Conventional technology	[81]	Skin diseases (*Fusarium oxysporum*, *T. rubrum* and *T. mentagrophytes*)	Seventeen EOs	EOs analysis by GC-MS and GC-FID. Antifungal activity by M38-A protocol. Cytotoxicity assay using MTT technique.	Strong activity was demonstrated in 70% to 80% of samples against dermatophytes and from 20% against *F. oxysporum*. The lowest MIC values were obtained with citral chemotype *Lippia alba* oil (BC2) at concentrations of 31.25 and 125 μg/mL on *T. rubrum* and *T. mentagrophytes*, respectively, but not against *F. oxysporum.* Oil from *Minthostachys mollis* (Kunth) Griseb showed strong activity against all fungi evaluated. Active samples against dermatophytes and *F. oxysporum* were not cytotoxic on vero cells ATCC CCL-81; excluding *Lippia origanoides* Kunth (5E), carvone chemotype Lippia alba (TS) and *M. mollis* oil (MEO2) oil. The EOs with the highest selectivity index (SI) values were *Aloysia triphylla* and *L. alba* oil (BC2) on dermatophytes. The main components of most active *L. alba* oils were characterized as carvone (TS, CC1) and citral (BC2). *To L. origanoides* oils, carvacrol (1A, 5E) and thymol (6F) were found as the main components. Pulegone and cis-piperitone epoxide were the main constituents of *M. mollis* MEO1 and MEO2 oils, respectively	The main components in these EOs may be responsible for their antifungal activity. This finding is very important, because it confirms the potential of these EOs as a source of new antidermatophytes.
25.	In vitro—Conventional technology	[82]	*Pityriasis versicolor* (*Malassezia furfur*)	*Citrus lemon* (lemon) and *Citrus sinensis* (orange)	Antifungal activity by disc diffusion and microdilution methods.	In screening lemon and orange oil by disc diffusion method, the diameter of inhibition zone was found to be 50 and 20 mm, i.e., greater than that of reference antibiotics, that is, gentamycin and streptomycin, which showed 16.5 and 17 mm, respectively. MIC of lemon and orange oil against *M. furfur* was found to be 0.8 and 2.2 µL/mL.	The findings in this study support the use of lemon and orange oil as a traditional herbal medicine for the control of *P. versicolor* infection in the skin.
26.		[83]	Skin infections (*C. albicans, C. dubliniensis, C. glabrata, C. guilliermondii, C. parapsilosis, C. tropicalis, C. neoformans, Sacharomyces cerevisiae, A. fumigatus,* *A. flavus, A. niger*, *T. rubrum, M. canis, Fonsecaea pedrosoi, Pseudallescheria boydii, Fusarium solani, S. schenckii*)	*Tropaeolum pentaphyllum* Lam. *tubers*	EO analysis by GC-MS. Antifungal activity by MIC.	GC-MS analysis revealed that the major EO constituent is benzyl isothiocyanate. The strongest effects against *Candida* spp. and dermatophytes were observed for the EO, with MIC well below 200 µg/mL.	Overall, the results support the use of the plant for the treatment of skin infections, and reveal the main active compounds.
27.	In vitro—Conventional technology	[84]	Skin infections (*T. rubrum* and *C. albicans*)	Twenty-nine oils	Antifungal activity by agar-well diffusion and MIC method.	Most of the EOs showed relatively high antimicrobial activity against all the tested organisms. The maximum antimicrobial activity was shown by *Calotropis gigantea*, followed by *Semecarpus anacardium, Azadirachta indica, Datura stromium, Coriandrum sativum, Luffa acutangula, Momordica cymbalaria, Gliricidia sepium, Hyptis sauveolens* and *O. sanctum* against the tested fungi. *C. gigantae* showed good antimicrobial activity against tested fungi, with MIC values ranging from 0.62 to 40 mg/mL, determined using inhibitory zone estimation.	The obtained results suggest that *C. gigantae* has antimicrobial activity. These results support the hypothesis that the plant oil can be used to cure skin diseases, and may have a role as a pharmaceutical compound and as a preservative.
28.	In vitro—Nonconventional technology	[85]	Skin infections (*Candida pseudotropicalis*)	*Lippia multiflora* (lippia oil)	Different concentrations of the oil were incorporated into six different ointment bases. Tween 80 was incorporated at different concentrations into ointments containing 10% *w/w* lippia oil; a concentration at which minimum antimicrobial activity was observed in the ointment bases. The viscosity, spreadability and stability of the formulations were determined. Antimicrobial activities of lippia oil in the formulations were determined against selected bacteria and fungi using the agar diffusion assay method; 10% *w/w* salicylic acid formulations were used as reference products.	Incorporation of lippia oil into the ointment bases led to reduction in their viscosity with increase in spreadability. None of the formulations showed antimicrobial activity at lippia oil content ≤ 5% *w/w*. Inclusion of Tween 80 in the formulations significantly increased the antimicrobial activities of the oil (*p* < 0.05). The antimicrobial activities of 10% *w/w* lippia oil formulated in absorption bases containing 6% *w/w* Tween 80 were significantly higher (*p* < 0.05) than those formulated in hydrophobic bases. Lippia oil ointment formulations showed greater antimicrobial activities than salicylic acid ointments. Two of the lippia oil ointment formulations bled when subjected to centrifugal force.	Ointment formulations of lippia oil (10% *w/w*) in absorption base (hydrous wool fat ointment BP or simple ointment BP) containing 6% *w/w* Tween 80 were found to be stable and very effective at inhibiting the growth of selected pathogens implicated in skin infections.
29.	In vitro—Nonconventional technology	[56]	Different skin infecting pathogens (*C. albicans, A. niger*)	Piper betle	Agar well disc diffusion method for zone of inhibition and broth dilution method for MIC. Gels were formulated using different polymers like hydroxy propyl methyl cellulos (HPMC), Carbopol 934, sodium carboxy methyl cellulose (sodium CMC), and sodium alginate. Formulation was evaluated for various physico-chemical parameters like pH, viscosity, consistency, homogeneity, spreadability, skin irritation test and stability studies.	*C. albicans* showed maximum sensitivity to the oil that was 24 mm at 20 μL and 20 mm at 20 μL for *A. niger*. Zone of inhibition of oil was not much affected by incorporation into a gel, as confirmed from the antimicrobial results of the final formulations. DMSO as the cosolvent for EO and carbopol 934 (1%), HPMC (5%) as gelling agent showed the best results in the final formulations.	The gel showed promising antifungal activity against other strains used in this study. The gel was stable at room temperature.
30.		[86]	Candidiasis (*C. krusei* ATCC 6528, *C. parapsilosis* ATCC 22019, *C. krusei, C. parapsilosis, C. albicans, C. glabrata, C. tropicalis*	*Bidens tripartita*	EO analysis by GC and GC-MS techniques. A hydrogel, bigel and oleogel were prepared using an RZR 2020 mechanical stirrer The antifungal properties of *B. tripartita* EO were also tested against *Candida* species using the modified agar-well diffusion method.	The therapeutic efficacy of the topical formulations is influenced by active phytoconstituents and vehicle characteristics. The highest activity was observed against *C. tropicalis* and *C. krusei* (3.7 mm difference, compared to hydrogel without BTEO). The main components were *p*-cymen (4.82%), β-linalool (2.85%).	Among the tested gel formulations combined with BTEO, the hydrogel-based formulation appeared to be the optimal combination, exhibiting antifungal activity against all tested *Candida* strains. This oil appeared to be a promising topical anticandidal agent.
31.		[87]	Candidiasis (*C. albicans* ATCC 10231, *C. krusei* ATCC 6258 and *C. parapsilosis* ATCC 90098)	Mediterranean EOs (*Rosmarinus officinalis, Lavandula x intermedia “Sumian*”, *Origanum vulgare* subsp. *hirtum*)	The antifungal susceptibility test for the selected (clotrimazole- nanostructured lipid carriers) CLZ-NLC formulations was performed based on the broth microdilution method for yeasts, with minor adaptations. For comparative purposes, in addition to the CLZ-LNLC (*Lavandula*) and CLZ-RNLC (*Rosmarinus*) formulations, LNLC and RNLC systems were also tested, as well as free EOs (L and R) and free CLZ. The cytotoxicity of all samples (pure EOs and NLC systems) was evaluated through the Alamar Blue^®^ method.	Results of the in vitro biosafety on HaCaT (normal cell line) and A431 (tumoral cell line) allowed us to select *Lavandula* and *Rosmarinus* as antiproliferative agents with the potential for use as coadjuvants in the treatment of nontumoral proliferative dermal diseases Nanoparticles provided prolonged in vitro release of clotrimazole. In vitro studies against *C. albicans*, *C. krusei* and *C. parapsilosis*, showed an increase of the antifungal activity of clotrimazole-loaded nanoparticles prepared with *Lavandula* or *Rosmarinus*, thus confirming that NLC-containing Mediterranean EOs represent a promising strategy to improve drug effectiveness against topical candidiasis.	Taken together, the results of this study imply that the nanoencapsulation of the selected antifungal drug into NLC systems prepared using Mediterranean EOs as intrinsic oily components is a promising strategy to improve CLZ effectiveness against candidiasis. In particular, the results open the debate concerning the possible application of the properties of *Lavandula* and *Rosmarinus*, whose synergistic effects must be further investigated; this could bring about a strategy for the use of CLZ-EO-NLC to overcome the drug resistance mechanisms associated with the treatment of topical infections.
32.	In vitro—Nonconventional technology	[88]	Skin infections due to pathogenic fungi(*C. albicans, C. tropicalis* and *T. mentagrophytes*)	Black pepper, Cardamom, Cumin, Boswellia and Patcholi	Antifungal activity by agar well diffusion method. For synergistic activity between Fluconazole and EO, sterile strips of 5 cm × 1 cm were used.	The results indicated that all the oils inhibited fungal strains in varying degrees of dilutions. Boswellia EO was found to be the most effective in terms of antifungal activity against *C. tropicalis* and Cardamom EO against *T. mentagrophytes*. The results indicated that Boswellia EO and Fluconazole in combination acted as the most powerful antifungal agent against *C. tropicalis*, even at 1:10 dilution and 100 μg/mL, respectively.	These results indicated that the active components present in EOs should be a focus area of future in vivo research, especially in conjunction with existing antifungal drugs. The molecular mechanisms, mode of action, stability, toxicity, and efficacy of the active components isolated from EOs need to be further studied and evaluated.
33.	In vitro and In vivo—Conventional technology	[89]	Body aches and dermatitis (*A. flavus*, *A. niger*, *F. solani*, *Mucor* piriformis, *Wickerhamomyces anomalus*, *Wickerhamomyces anomalus*, *Deboromyces hansenii,* and *C. albicans*)	*Parrotiopsis jacquemontiana*	Phytochemical characterization of oil was determined using GC-MS and Fourier Transformed- Infrared spectroscopy (FT-IR). The antimicrobial potential of oil was investigated by disc diffusion method. Wound healing was performed in vivo with determination of wound contraction rates, histopathology, hemostatic potential and hydroxyproline estimation.	GC-MS analysis indicated that the oil was constituted mainly of 2, 6-dimethyl-8-oxoocta-2, 6-dienoic acid, methyl ester (18.2%), syringol (17.8%), catechol (12.4%), guaiacol (5.2%), p-cresol (5.4%) and phenol, 2-propyl- (3.7%). FT-IR analysis revealed several important functional groups in its chemical composition, especially phenolic O-H compound stretching. The oil strongly inhibited the growth of various fungal isolates with low minimum inhibitory concentrations (64–256) μg/mL. Remarkable rate for wound closure and epithelization, hemostatic potential and marked increase (*p* < 0.05) in hydroxyproline content were observed during wound healing in rat.	The results suggested that oil can be used as a potential source of wound healing therapeutics.
34.	In vitro and In vivo—Conventional technology	[90]	Skin infections (*Chrysosporium tropicum, Trichophyton simii, T. rubrum* and *Chrysosporium indicum*)	*Trachyspermum ammi*	EO analysis by GC-MS. Antidermatophytic activity was determined by disc diffusion method and minimum inhibitory concentration. Acute dermal irritation assay was applied for toxicological studies on albino mice.	Thymol was found to be the major compound (58.88%), followed by p-cymene (24.02%), γ-terpinene (13.77%) and β-pinene (1.90%). Maximum zone of inhibition was observed against *C. tropicum* (63.83 +/− 0.166 mm) followed by *T. simii* (57 +/− 0.288 mm), *T. rubrum* (51.33 +/− 0.333 mm) and *C. indicum* (45 +/− 0.577 mm). Five fractions were separated under different temperature conditions, labelled as TA(I)-TA(V). Maximum effects were seen in case of TA(IV) and TA(V) fractions. Excellent results of TA(V) were observed against *M. gypseum* (0.015 +/− 0.002 mu l/mL), followed by *M. canis* (0.017 +/− 0.002 mu l/mL), *T. rubrum* (0.02 +/− 0.000 mu l/mL) and *C. albicans* (0.05 +/− 0.003 mu l/mL). Low concentrations, i.e., up to 3%, did not show any irritation on mice skin. At 5%, three mice showed mild erythema, while at 7% concentration, all five mice exhibited well-defined erythema	This study concluded that the EO of *T. ammi* and its fractions have strong antidermatophytic properties with no side effect at low concentrations, and thus, could represent alternatives to antibiotics.
35.		[91]	Skin disease due to fungi strains(*C. albicans, C. parapsilosis, C. tropicalis; A. niger, A. terreus, A. flavus, A. fumigatus, Penicillium* sp. and *Mucor* sp.	Lemon grass (*Cymbopogon citratus* (DC.) Stapf)	LGEO chemical profiling by GC-MS. Antifungal activity using disc diffusion and vapor diffusion methods. The anti-inflammatory potential of LGEO was assessed by the carrageenan-induced paw edema test. Topical anti-inflammatory activity was evaluated as inhibition of croton oil-induced ear edema in mice. The resulting inflammatory reaction was also evaluated by microscopic inspection.	The major components were geranial (42.2%), neral (31.5%), and β-myrcene (7.5%). LGEO exhibited promising antifungal effects against *C. albicans, C. tropicalis*, and *A. niger*, with different inhibition zone diameters (IZDs) (35–90 mm). IZD increased with increasing oil volume. Higher anti-*Candida* activity was observed in the vapor phase. In an evaluation of the anti-inflammatory effect, LGEO (10 mg/kg, administered orally) was shown to significantly reduce carrageenan-induced paw edema, with a similar effect to that of oral diclofenac (50 mg/kg), which was used as the positive control. Oral administration of LGEO showed dose-dependent anti-inflammatory activity. In addition, topical application of LGEO in vivo indicated a potent anti-inflammatory effect, as demonstrated by the use of a mouse model of croton oil-induced ear edema. To our knowledge, this is the first such report to be published. The topical application of LGEO at doses of 5 and 10 µL/ear significantly reduced acute ear edema induced by croton oil in 62.5 and 75% of mice, respectively. In addition, histological analysis clearly confirmed that LGEO inhibits the skin inflammatory response in animal models.	The results of this present study indicated that LGEO has potential in the development of drugs for the treatment of fungal infections and skin inflammation that should be explored in future studies.
36.	In vitro and In vivo—Conventional technology	[92]	Skin infections (*M. gypseum* and *T. mentagrophytes*)	*Ageratum houstonianum* Mill.,	The proportions of EO/griseofulvin tested were 8:2 and 10:1 (*w/w*). The acute dermal toxicity of this oil was evaluated on guinea-pigs (*Cavia porcellus*) using a standard method. Agar dilution method with serial dilution of the oil and the mixtures was used for antidermatophytic tests.	The MIC of the EO was 80 μg/mL for the tested dermatophytes. The 8:2 mixture was 20 μg/mL, whereas those of the 10:1 mixture were 8 and 10 ug mL for *M. gypseum* and *T. mentagrophytes*, respectively. Compared to the control group, we noted no diarrhoea, nor any change in treated skin or fur appearance. In contrast, the degree of sensitivity to noise, reaction to pinch, activity (locomotion) and reactivity decreased with increases in the dose. The LD50 was determined to be 5 g/kg b.wt.	These data suggest that the EO of the leaves of *A. houstonianum* contains antidermatophytic compounds, and may not be toxic when used topically. A potentialization effect was observed between the EO and griseofulvin.
37.	In vitro and In vivo—Nonconventional technology	[93]	Skin disease (*C. albicans* strain ATC 100231)	Lemongrass oil (LGO) is a volatile oil extracted from the leaves of *C. citratus*	MIC and MFC of LGO were determined using the broth macrodilution method. LGO-loaded nanosponges were prepared using the emulsion solvent evaporation method. The irritation effect of the chosen gel formulation, F9, was evaluated by carrying out a Draize patch test on six albino rats. The animals’ backs were shaved 24 h before the application of the formulation, and then 0.5 g gel was spread uniformly on the hair-free skin within an area of 4 cm^2^. The skin was observed for any visible changes after 24, 48, and 72 h from the application of the formulation. Male albino rats weighing 150–180 g were housed in individual cages and given access to food and water ad libitum. Animals were subjected to intraperitoneal injection of cyclophosphamide (100 mg/kg, body weight) for three days before fungal infection to suppress their immunity, with the aim of achieving a heavy cutaneous infection. *C. albicans*, ATCC 10231 was used to induce infection.	The minimal inhibitory concentration and minimal fungicidal concentration of LGO against *C. albicans* strain ATC 100231, determined using the broth macrodilution method were found to be 2 and 8 µL/mL, respectively. The selected formulation, F9, was tested for skin irritation and antifungalactivity against *C. albicans*; the results confirmed the nonirritancy and effective antifungal activity of the prepared hydrogel.	LGO was successfully incorporated into an EC nanosponge using the emulsion solvent evaporation method, followed by integration into a carbopol hydrogel. Among the nine prepared hydrogels integrating lemongrass-loaded nanosponges, the F9 formulation was chosen for further study on the basis of its particle size and controlled release profile. The selected formula showed no skin irritation and in vivo antifungal activity in male albino rats. These results are promising with respect to the practical application of the incorporation of LGO in pharmaceutical formulations with the benefit of decreasing the hazards of its use in folk medicine in crude form.
38.	In vitro and In vivo—Nonconventional technology	[94]	Skin infections (*T. mentagrophytes, T. rubrum, T. verrucosum* and *M. canis*)	*Cymbopogon martini*	Hydrodistillation aerial part *C. martini* and topical formulations were prepared in five different semisolid bases. In vitro antimicrobial investigations were performed on EO and topical formulations. Skin sensitizations of the formulations were evaluated using guinea pig maximization-	*C. martini* EO has shown broad-spectrum antimicrobial potency against all tested organisms, with MIC value ranging from 0.65 to 10 μg/mL. Absolute inhibitions of growth of fungi were observed against *T. mentagrophytes* and *T. rubrum* at concentrations above 1% of oil and against *M. canis* and *T. verrucosum* at a concentration of 4% oil. Among topical formulations, the highest antimicrobial activity was recorded in hydrophilic ointment, followed by the macrogol blend ointment. The antimicrobial activity of oil was higher in fungal pathogens compared to bacteria. Gram-positive bacteria were more sensitive than gram-negative bacteria. A hydrophilic and macrogol blend containing 5% oil did not produce any skin sensitization on guinea pigs.	Topical formulations of *C. martini* EO may be alternative topical agents with safe broad-spectrum activity for the treatment of skin disorders. Further studies should focus on shelf-life and clinical studies of the product.
39.	In vitro and In vivo—Nonconventional technology	[95]	Dermatophytes and some filamentous fungi *(A. flavus, A. niger, A. fumigatus, Microspoum audouni, M. nanum, T. mentagrophytes, T. verrucosm* and *T. violaceum*)	*C. martini* and *Chenopodium ambrosioides*	Chemical constituents of EOs were determined by GC–MS. The fungitoxicity of EOs and their combinations was also evaluated against some filamentous fungi and other species of dermatophytes by poisoned food technique at concentrations of 200, 600 and 800 ppm. Male guinea pigs (5–8 months old, 350–400 g) were obtained from Central Animal House, Institute of Medical Sciences, BHU, Varanasi. The ointment of each EO and their combination were prepared by mixing 1 mL of the oil with 100 g petroleum jelly. Treatments were started from the sixth day after inoculation and continued until complete recovery was achieved. To the infected area was applied 0.2 g ointment twice a day in the treatment set. In the control set, only petroleum jelly without EO was used.	The major constituents of the essential oil of *C. martini* were *trans*-geraniol (60.9%), β-elemene (12.26%), *E*-citral (3.95%) and linalool (3.44%). The minimum inhibitory concentrations of EO and of their combination were found between 150 and 500 ppm, while those of known antifungal drugs ranged from 1000 to 5500 ppm. EOs ointments were prepared and applied against induced ringworm in a guinea pig model. Disease removal was observed in 7–21 days, and the hair samples showed negative results for fungal culture in a time-dependent manner after the application of EO ointments.	The results provided scientific validation for the use of these EOs in the treatment of dermatophyte infections; as such, the topical application of these EOs may be recommended as an alternative to synthetic drugs.
40.	Clinical Interventions	[96]	Ringworm (*tenia corporis*) (*M.* and *T. mentagrophytes)*	*Curcuma longa* L.	The oil was formulated in the form of an ointment, 1% w/v and subjected to topical testing on patients of the Out Patient Department (OPD) at Moti Lal Nehru Medical College, Allahabad. Patients were selected on the basis of potassium hydroxide (KOH) positive results and diagnosis of tenia corporis.	A *C. longa* leaves EO was fungicidal at 2.5 μL/mL, at which it was tolerated in heavy doses. The fungicidal activity was thermostable up to 80 °C and a shelf life of up to 24 months was determined. The oil also showed a broad fungitoxic spectrum, inhibiting the mycelial growth of other fungi, viz., *E. floccosum, M. nanum, T. rubrum, T. violaceum*. Moreover, at up to 5% concentration, it did not exhibit any adverse effect on mammalian skins. After the second week of treatment, all patients were KOH-negative. At the end of medication, 75% of patients recovered completely, while 15% showed significant improvement from the disease.	After ongoing successful clinical trials, it was concluded that the ointment could be exploited commercially. The relationship of dermatophytes to the toxicity of the oil, vis-a-vis phylogeny, evaluated using the molecular data of the pathogens, was also discussed.
41.		[97]	*Pityriasis versicolor* (*Malassezia* sp.)	*Myrtus communis*	Forty one patients with PV that had not received any treatment for PV two weeks prior to enrollment. Samples were taken from four different locations: the surface layers of the back, thorax, arms and facial skin (area of 4 cm^2^), using adhesive tape. Chemical profiling via LC–MS/MS. Antifungal effect of EO on *Malassezia* strains was tested using European committee on antimicrobial susceptibility testing (EUCAST) broth microdilution method.	Study was performed on 22 men (~ 54%) and 19 women (~ 46%) aged 20–80 years (mean 43 ± 16). A total of 86 yeast colonies were isolated from 41 patients with PV. Seven different *Malassezia* species were identified as follows: *M. furfur* (42.5%, 37/86), *M. sympodialis* (23.5%, 20/86), *M. slloofae* (13.9%, 12/86), *M. globosa* (7.5%, 7/86), M. obtusa (6%, 5/86), *M. japonica* (4%, 3/86) and *M. restricta* (2.5%, 2/86). The most and least infected sites were back (74.5, 31/41) and forehead (11.8%, 5/41), respectively. Inhibition of fungal growth with *M. communis* EO was efective in 96% (36/37) of isolated *M. furfur*, 83% (16/20) of *M. sympodialis*, 78% (9/12) of M. sloofae, 78% (5/7) of *M. globosa*, 75% (4/5) *M. obtusa*, 73% (2/3) of *M. japonica* and 62% (1/2) of *M. restricta*.	The broad-spectrum antimicrobial activities of *M. communis* EO and its potent inhibiting activity on *Malassezia* growth deserve further research, with the aim of potentially applying this EO in the topical treatment of skin diseases

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
