# Peer review of "Antifungal Properties of Essential Oils and Their Compounds for Application in Skin Fungal Infections: Conventional and Nonconventional Approaches"

_molecules, 2021, doi:10.3390/molecules26041093_

Round 1

Reviewer 1 Report

The paper is important because fungal infections and the chemoresistance of fungal strains are difficult problems nowadays. The authors collected a large number of data regarding EO usage in fungal infections. This is nice but the criticism of the applied  methods is missing. EOs are very hydrophobic and most of the classical methods use solubilization agents or some other techniques to overcome of the difficulties of the antifungal tests that are done mostly in aqueous environment. 

The authors do not cite papers that are dealig with novel formulations (e.g. cyclodextrin entrapment, Pickering type nanoemulsions).

It would be useful to add a paragraph on the shortages of the classical methods and the advanatages of the novel formulations.

There are misspelling mistakes in teh text that should be corrected.

Author Response

Point 1:

The authors do not cite papers that are dealig with novel formulations (e.g. cyclodextrin entrapment, Pickering type nanoemulsions).

It would be useful to add a paragraph on the shortages of the classical methods and the advanatages of the novel formulations.

Response 1: A section has been added to include studies on cyclodextrins, pickering type nanoemulsions and advantages of non-conventional techniques (line 739-770).

Point 2: There are misspelling mistakes in teh text that should be corrected.

Response 2: Corrections have been made for spelling mistakes in the text.  

Reviewer 2 Report

Reviewed article is good written and very interesting. First Authors presented fungal pathogens and drugs used in their treatment. Next are descriptions of essential oils and their antifungal activity. In results are included antifungal activity of essential oils, their main active compounds, and used methods. In Table are presented about 40 studies of antifungal essential oils effective in human skin diseases. Authors wrote also very extensive Discussion. I have only some small points for correction:

  1. In Abstract are mainly information about methodology. I suggest remove some sentences and input results wit the most important and active essential oil and their main compounds.
  2. In Table 1 are for correction No., because are e.g. 1., 2.4, 3.6, 4.7, instead of 1., 2., 3., 4., etc. In Findings, I suggest mention at least 2 main chemical compounds of essential oils.
  3. In Discussion please add and cite other recent review entitled "Essential Oils of Lamiaceae Family Plants as Antifungals" https://www.mdpi.com/2218-273X/10/1/103

Author Response

Point 1: In Abstract are mainly information about methodology. I suggest remove some sentences and input results wit the most important and active essential oil and their main compounds.

Response 1: The abstract has been revised accordingly. The most active compounds and the plant species that portray the excellent antifungal properties have been included in the abstract.

Point 2: In Table 1 are for correction No., because are e.g. 1., 2.4, 3.6, 4.7, instead of 1., 2., 3., 4., etc. In Findings, I suggest mention at least 2 main chemical compounds of essential oils.

Response 2: We have formatted Table 1 on the whole (font and alignment) for better clarity. The numbering format has been corrected. We have included the main chemical compounds (for all the studies that involved the determination of EO compositions in their study in Findings).

Point 3: In Discussion please add and cite other recent review entitled "Essential Oils of Lamiaceae Family Plants as Antifungals" https://www.mdpi.com/2218-273X/10/1/103

Response 3: The suggested reference has been added (line 399-405).

Round 2

Reviewer 1 Report

The paper's quality has been improved especially by introducing the latest approaches to solubilize EOs.

The workk handles a lot of experimental approaches in assessment of the EOs biological actions.

What I still miss is the criticisms of the so called conventional methods because in these approaches solubilizers are appplied (or not) which can falsify the results.

Author Response

Point 1: What I still miss is the criticisms of the so called conventional methods because in these approaches solubilizers are appplied (or not) which can falsify the results.

Response 1: We have included the criticism in the manuscript (line 581-600). We hope this answers the questions.